# Generative AI enables medical image segmentation in ultra low-data regimes

Li Zhang[1], Basu Jindal[1], Ahmed Alaa [2,3], Robert Weinreb [4], David Wilson [5], Eran Segal [6,7], James Zou [8,9] & Pengtao Xie [1,10] ✉

Semantic segmentation of medical images is pivotal in applications like disease diagnosis and treatment planning. While deep learning automates this task effectively, it struggles in ultra low-data regimes for the scarcity of annotated segmentation masks. To address this, we propose a generative deep learning framework that produces high-quality image-mask pairs as auxiliary training data. Unlike traditional generative models that separate data generation from model training, ours uses multi-level optimization for end-to-end data generation. This allows segmentation performance to guide the generation process, producing data tailored to improve segmentation outcomes. Our method demonstrates strong generalization across 11 medical image segmentation tasks and 19 datasets, covering various diseases, organs, and modalities. It improves performance by 10–20% (absolute) in both same- and out-of-domain settings and requires 8–20 times less training data than existing approaches. This greatly enhances the feasibility and cost-effectiveness of deep learning in data-limited medical imaging scenarios.

Medical image semantic segmentation[1–3] is a pivotal process in the modern healthcare landscape, playing an indispensable role in diagnosing diseases[4], tracking disease progression[5], planning treatments[6], assisting surgeries[7], and supporting numerous other clinical activities[8,9]. This process involves classifying each pixel within a specific image, such as a skin dermoscopy image, with a corresponding semantic label, such as skin cancer or normal skin.

The advent of deep learning has revolutionized this domain, offering unparalleled precision and automation in the segmentation of medical images[1,2,10,11]. Despite these advancements, training accurate and robust deep learning models requires extensive, annotated medical imaging datasets, which are notoriously difficult to obtain[9,12]. Labeling semantic segmentation masks for medical images is both time-intensive and costly, as it necessitates annotating each pixel. It requires not only substantial human resources but also specialized domain expertise. This leads to what is termed as ultra low-data regimes—scenarios where the availability of annotated training images is remarkably scarce. This scarcity poses a substantial challenge to the existing deep learning methodologies, causing them to overfit to training data and exhibit poor generalization performance on test images.

To address the scarcity of labeled image-mask pairs in semantic segmentation, several strategies have been devised, including data augmentation and semi-supervised learning approaches. Data augmentation techniques[13–16] create synthetic pairs of images and masks, which are then utilized as supplementary training data. A significant

[1]Department of Electrical and Computer Engineering, University of California San Diego, La Jolla, CA, USA. [2]Bakar Computational Health Sciences Institute, University of California San Francisco, San Francisco, CA, USA. [3]Department of Electrical Engineering and Computer Sciences, University of California Berkeley, Berkeley, CA, USA. [4]Hamilton Glaucoma Center, Shiley Eye Institute, Viterbi Family Department of Ophthalmology, University of California San Diego, La Jolla, CA, USA. [5]Division of Pulmonary, Allergy and Critical Care Medicine, Department of Medicine, University of Pittsburgh, Pittsburgh, PA, USA. [6]Department of Computer Science and Applied Mathematics, Weizmann Institute of Science, Rehovot, Israel. [7]Department of Molecular Cell Biology, Weizmann Institute of Science, Rehovot, Israel. [8]Department of Biomedical Data Science, Stanford University School of Medicine, Stanford, CA, USA. [9]Department of Computer Science, Stanford University, Stanford, CA, USA. [10]Department of Medicine, University of California San Diego, La Jolla, CA, USA. ✉e-mail: p1xie@ucsd.edu

limitation of these methods is that they treat data augmentation and segmentation model training as separate activities. Consequently, the process of data augmentation is not influenced by segmentation performance, leading to a situation where the augmented data might not contribute effectively to enhancing the model's segmentation capabilities. Semi-supervised learning techniques[8,17–20] exploit additional, unlabeled images to bolster segmentation accuracy. Despite their potential, these methods face limitations due to the necessity for extensive volumes of unlabeled images, a requirement often difficult to fulfill in medical settings where even unlabeled data can be challenging to obtain due to privacy issues, regulatory hurdles (e.g., IRB approvals), among others. Recent advancements in generative deep learning[21–23] have opened new possibilities for overcoming such challenges by generating synthetic data. Compared to traditional augmentation methods, generative models have the potential to produce more realistic and diverse samples. However, most existing data generation or augmentation approaches[13–16] do not incorporate feedback from the segmentation performance itself. Some recent studies[24] have proposed multi-level optimization (MLO) frameworks in which the data generation process is guided by downstream tasks, such as classification. Yet, applying such optimization effectively to segmentation tasks remains underexplored. Moreover, unlike semi-supervised segmentation methods[8,17–20], generative approaches have the advantage of not requiring additional unlabeled data—an important benefit in sensitive medical domains.

In this work, we introduce GenSeg, a generative deep learning framework designed to address the challenges of ultra low-data regimes in medical image segmentation. GenSeg generates high-fidelity paired segmentation masks and medical images through a MLO process directly guided by segmentation performance. This ensures that the generated data not only meets high-quality standards but is also optimized to improve downstream model training. Unlike existing augmentation methods, GenSeg performs end-to-end data generation tightly coupled with segmentation objectives; unlike semi-supervised approaches, it requires no additional unlabeled images. GenSeg is a versatile, model-agnostic framework that can be seamlessly integrated into existing segmentation pipelines. We validated GenSeg across 11 segmentation tasks and 19 datasets spanning diverse imaging modalities, diseases, and organs. When integrated with UNet[1] and DeepLab[10], GenSeg significantly boosts performance in ultra low-data settings (e.g., using only 50 training examples), achieving absolute gains of 10–20% in both same-domain and out-of-domain (OOD) generalization. Additionally, GenSeg demonstrates strong data efficiency, matching or exceeding baseline performance while requiring 8–20 × fewer labeled samples.

## Results

### GenSeg overview

GenSeg is an end-to-end data generation framework designed to generate high-quality, labeled data, to enable the training of accurate medical image segmentation models in ultra low-data regimes (Fig. 1a). Our framework integrates two components: a data generation model and a semantic segmentation model. The data generation model is responsible for generating synthetic pairs of medical images and their corresponding segmentation masks. This generated data serves as the training material for the segmentation model. In our data generation process, we introduce a reverse generation mechanism. This mechanism initially generates segmentation masks, and subsequently, medical images, adhering to a progression from simpler to more complex tasks. Specifically, given an expert-annotated real segmentation mask, we apply basic image augmentation operations to produce an augmented mask, which is then inputted into a deep generative model to generate the corresponding medical image. A key distinction of our method lies in the architecture of this generative model. Unlike traditional models[22,23,25,26] that rely on manually

designed architecture, our model automatically learns this architecture from data (Fig. 1b, c). This adaptive architecture enables more nuanced and effective generation of medical images, tailored to the specific characteristics of the augmented segmentation masks.

GenSeg features an end-to-end data generation strategy, which ensures a synergistic relationship between the generation of data and the performance of the segmentation model. By closely aligning the data generation process with the needs and feedback of the segmentation model, GenSeg ensures the relevance and utility of the generated data for effective training of the segmentation model. To evaluate the effectiveness of the generated data, we first train a semantic segmentation model using this data. We then assess the model's performance on a validation set consisting of real medical images, each accompanied by an expert-annotated segmentation mask. The model's validation performance serves as a reflection of the quality of the generated data: if the data is of low quality, the segmentation model trained on it will show poor performance during validation. By concentrating on improving the model's validation performance, we can, in turn, enhance the quality of the generated data.

Our approach utilizes a MLO[24] strategy to achieve end-to-end data generation. MLO involves a series of nested optimization problems, where the optimal parameters from one level serve as inputs for the objective function at the next level. Conversely, parameters that are not yet optimized at a higher level are fed back as inputs to lower levels. This yields a dynamic, iterative process that solves optimization problems in different levels jointly. Our method employs a three-tiered MLO process, executed end-to-end. The first level focuses on training the weight parameters of our data generation model, while keeping its learnable architecture constant. This training is performed within a generative adversarial network (GAN) framework[22] (Fig. 1d), where a discriminator network learns to distinguish between real and generated images, and the data generation model is optimized to fool the discriminator by producing images that closely resemble real ones. At the second level, this trained model is used to produce synthetic image-mask pairs, which are then employed to train a semantic segmentation model. The final level involves validating the segmentation model using real medical images with expert-annotated masks. The performance of the segmentation model in this validation phase is a function of the architecture of the generation model. We optimize this architecture by minimizing the validation loss. By jointly solving the three levels of nested optimization problems, we can concurrently train data generation and semantic segmentation models in an end-to-end manner.

Our framework was validated for a variety of medical imaging segmentation tasks across 19 datasets, spanning a diverse spectrum of imaging techniques, diseases, lesions, and organs. These tasks comprise segmentation of skin lesions from dermoscopy images, breast cancer from ultrasound images, placental vessels from fetoscopic images, polyps from colonoscopy images, foot ulcers from standard camera images, intraretinal cystoid fluid from optical coherence tomography (OCT) images, lungs from chest X-ray images, and left ventricles and myocardial wall from echocardiography images.

### GenSeg enables accurate segmentation in ultra-low data regimes

We evaluated GenSeg's performance in ultra-low data regimes. We conducted three independent runs for each dataset using different random seeds. The reported results represent the mean and standard deviation computed across these runs. GenSeg, being a versatile framework, facilitates training various backbone segmentation models with its generated data. To demonstrate this versatility, we applied GenSeg to two popular models: UNet[1] and DeepLab[10], resulting in GenSeg-UNet and GenSeg-DeepLab, respectively. GenSeg-DeepLab and GenSeg-UNet demonstrated significant performance improvements over DeepLab and UNet in scenarios with limited data (Fig. 2a

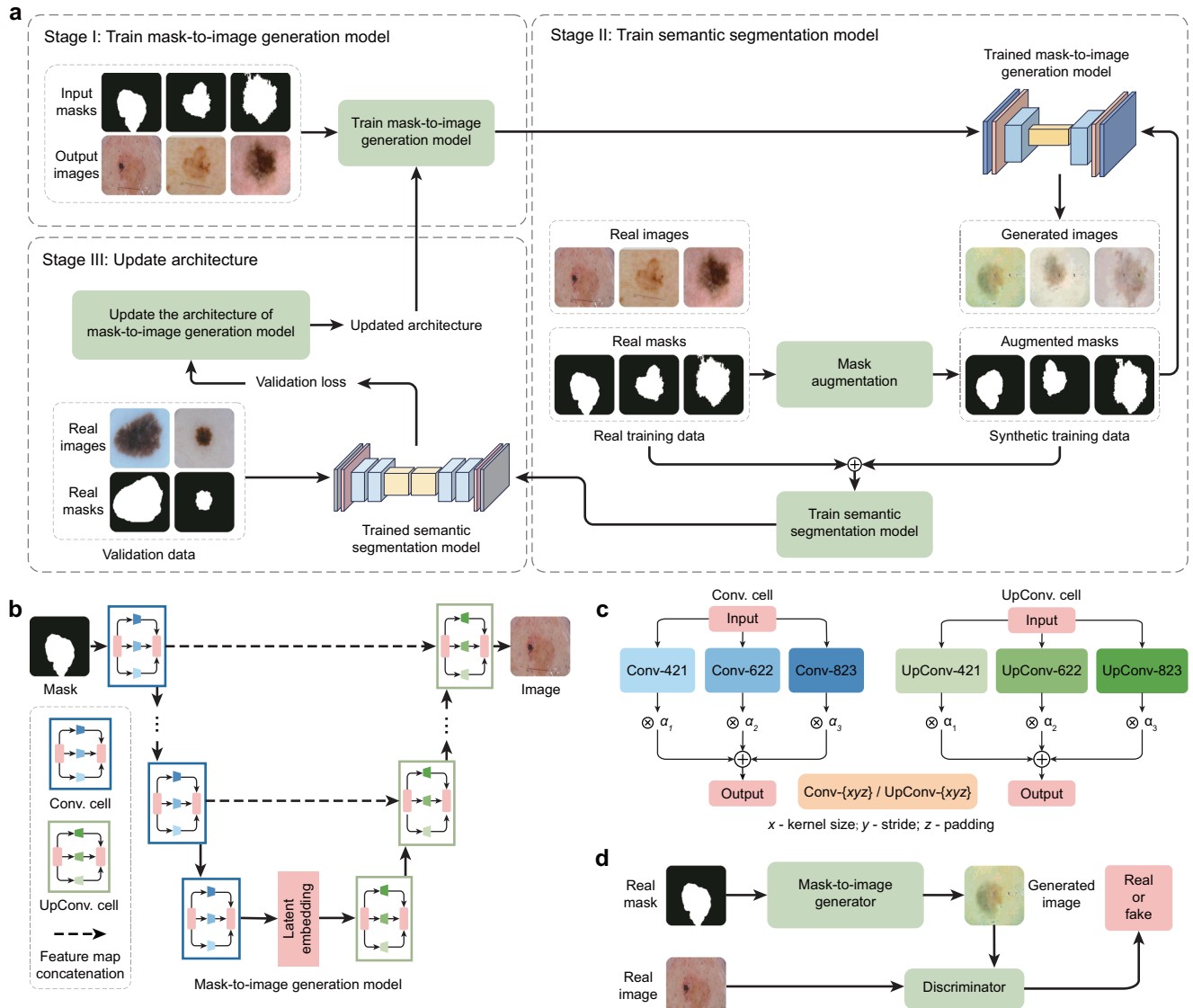

**Fig. 1 | Proposed end-to-end data generation framework for improving medical image segmentation in ultra low-data regimes. a** Overview of the GenSeg framework. GenSeg consists of (1) a semantic segmentation model that predicts a segmentation mask from an input image, and (2) a mask-to-image generation model that synthesizes an image from a segmentation mask. The latter features a neural architecture that is both learnable in structure and parameterized by trainable weights. GenSeg operates through three end-to-end learning stages. In stage I, the network weights of the mask-to-image model are trained with real mask-image pairs, with its architecture tentatively fixed. Stage II involves using the trained mask-to-image model to synthesize training data. Real segmentation masks are augmented to create new masks, from which synthetic images are generated. These synthetic image-mask pairs are used alongside real data to train the segmentation model. In stage III, the trained segmentation model is evaluated on a real validation dataset, and the resulting validation loss−which reflects the performance of the mask-to-image model−is used to update this architecture. Following

this update, the model re-enters Stage I for further training, and this cycle continues until convergence. **b** Searchable architecture of the mask-to-image generation model. It comprises an encoder and a decoder. The encoder processes an input mask into a latent representation using a series of searchable convolution (Conv.) cells. The decoder employs a stack of searchable up-convolution (UpConv.) cells to transform the latent representation into an output medical image. Each cell, as shown in (**c**) contains multiple candidate operations characterized by varying kernel sizes, strides, and padding options. Each operation is associated with a weight $\alpha$ denoting its importance. The architecture search process optimizes these weights, and only the most influential operations are retained in the final model. **d** The weight parameters of the mask-to-image generator are trained within a generative adversarial network (GAN) framework, in which a discriminator learns to distinguish real images from generated ones, while the generator is optimized to produce images that are indistinguishable from real images. All qualitative examples are sourced from publicly available datasets.

and Supplementary Fig. 1). Specifically, in the tasks of segmenting placental vessels, skin lesions, polyps, intraretinal cystoid fluids, foot ulcers, and breast cancer, with training sets as small as 50, 40, 40, 50, 50, and 100 samples respectively, GenSeg-DeepLab outperformed DeepLab substantially, with absolute percentage gains of 20.6%, 14.5%, 11.3%, 11.3%, 10.9%, and 10.4%. Similarly, GenSeg-UNet surpassed UNet by significant margins, recording absolute percentage improvements of 15%, 9.6%, 11%, 6.9%, 19%, and 12.6% across these tasks. The limited size of these training datasets presents significant challenges for

accurately training DeepLab and UNet models. For example, DeepLab's effectiveness in these tasks is limited, with performance varying from 0.31 to 0.62, averaging 0.51. In contrast, using our method, the performance significantly improves, ranging from 0.51 to 0.73 and averaging 0.64. This highlights the strong capability of our approach to achieve precise segmentation in ultra low-data regimes. Moreover, these segmentation tasks are highly diverse. For example, placental vessels involve complex branching structures, skin lesions vary in shape and size, and polyps require differentiation from surrounding

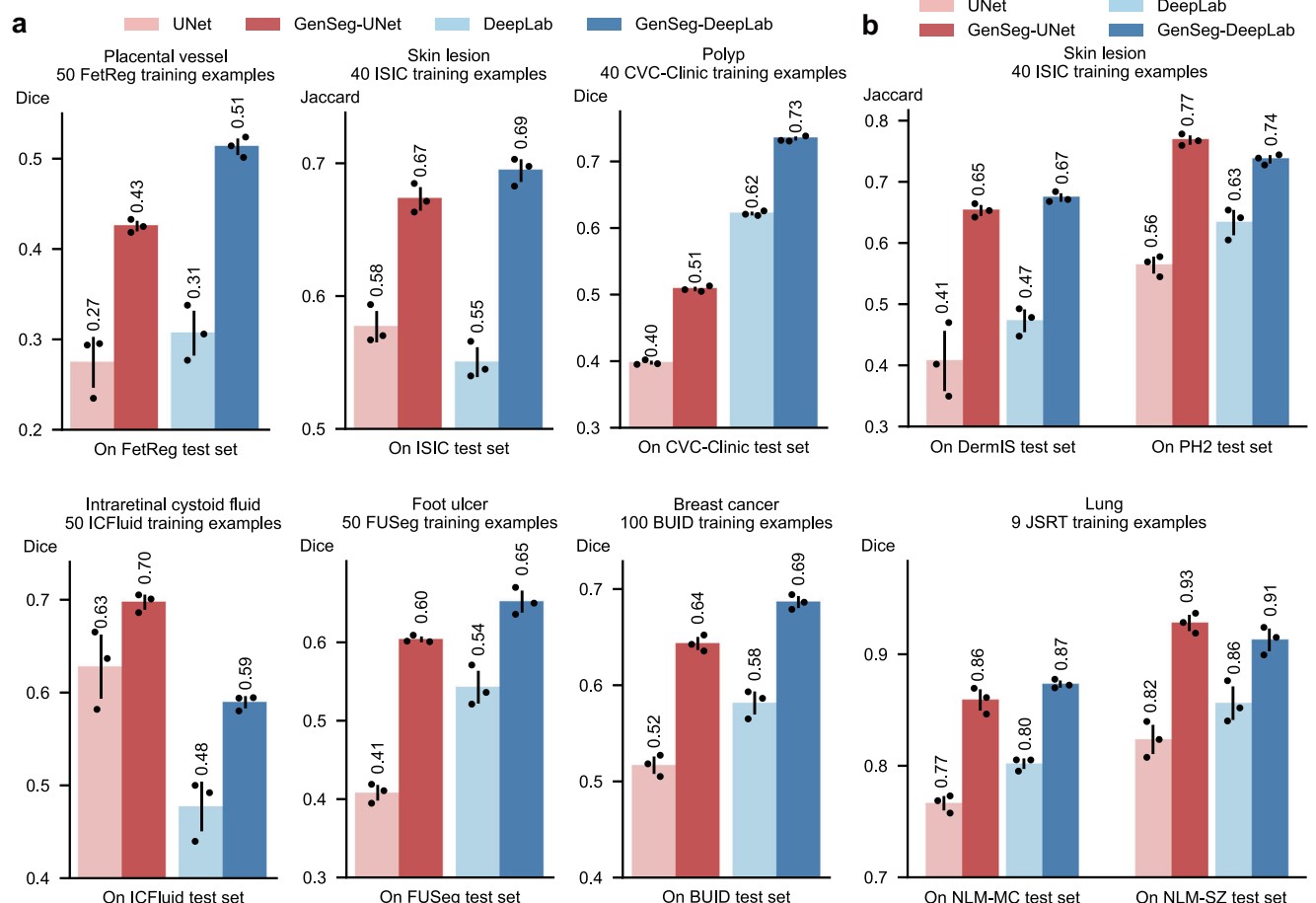

**Fig. 2 | GenSeg significantly boosted both in-domain and out-of-domain generalization performance, particularly in ultra low-data regimes. a** The performance of GenSeg applied to UNet (GenSeg-UNet) and DeepLab (GenSeg-DeepLab) under in-domain settings (test and training data are from the same domain) in the tasks of segmenting placental vessels, skin lesions, polyps, intraretinal cystoid fluids, foot ulcers, and breast cancer using limited training data (50, 40, 40, 50, 50, and 100 examples from the FetReg, ISIC, CVC-Clinic, ICFluid, FUSeg, and BUID datasets, respectively for each task), compared to vanilla UNet and DeepLab. **b** The performance of GenSeg-UNet and GenSeg-DeepLab under out-of-domain settings (test and training data are from different domains) in segmenting skin lesions (using only 40 examples from the ISIC dataset for training, and the DermIS and PH2 datasets for testing) and lungs (using only 9 examples from the JSRT dataset for training, and the NLM-MC and NLM-SZ datasets for testing), compared to vanilla UNet and DeepLab. In all panels, bar heights represent the mean, and error bars indicate the standard deviation across three independent runs with different random seeds. Results from individual runs are shown as dot points. Source data are provided as a Source Data file.

mucosal tissue. GenSeg demonstrated robust performance enhancements across these diverse tasks, underscoring its strong capability in achieving accurate segmentation across different diseases, organs, and imaging modalities.

**GenSeg enables robust generalization in out-of-domain settings**
Besides in-domain evaluation, where the test and training images were from disjoint subsets of the same dataset, we also evaluated GenSeg's effectiveness in OOD scenarios, wherein the training and test images originate from distinct datasets. The OOD evaluations were also conducted in ultra low-data regimes, where the number of training examples was restricted to only 9 or 40. Our evaluations focused on two segmentation tasks: the segmentation of skin lesions from dermoscopy images and the segmentation of lungs from chest X-rays. For the task of skin lesion segmentation, we trained our models using 40 examples from the ISIC dataset. These models were then tested on two external datasets, DermIS and PH2, to evaluate their performance outside the ISIC domain. In the lung segmentation task, we utilized 9 training examples from the JSRT dataset and conducted evaluations on two additional datasets, NLM-SZ and NLM-MC, to test the models' adaptability beyond the JSRT domain. GenSeg showed superior OOD generalization capabilities (Fig. 2b). In skin lesion segmentation,

GenSeg-UNet substantially outperformed UNet, achieving a Jaccard index of 0.65 compared to UNet's 0.41 on the DermIS dataset, and 0.77 versus 0.56 on PH2. Similarly, in lung segmentation, GenSeg-UNet demonstrated superior performance with a Dice score of 0.86 compared to UNet's 0.77 on NLM-MC, and 0.93 against 0.82 on NLM-SZ. Similarly, GenSeg-DeepLab significantly outperformed DeepLab: it achieved 0.67 compared to 0.47 on DermIS, 0.74 vs. 0.63 on PH2, 0.87 vs. 0.80 on NLM-MC, and 0.91 vs. 0.86 on NLM-SZ. Figure 3 and Supplementary Fig. 7 visualize some randomly selected segmentation examples. Both GenSeg-UNet and GenSeg-DeepLab accurately segmented a wide range of disease targets and organs across various imaging modalities with their predicted masks closely resembling the ground truth, under both in-domain (Fig. 3a and Supplementary Fig. 7) and OOD (Fig. 3b) settings. In contrast, UNet and DeepLab struggled to achieve similar levels of accuracy, often producing masks that were less precise and exhibited inconsistencies in complex anatomical regions. This disparity underscores the advanced capabilities of GenSeg in handling varied and challenging segmentation tasks. Supplementary Fig. 8 presents several mask-image pairs generated by GenSeg. The generated images not only exhibit a high degree of realism but also demonstrate excellent semantic alignment with their corresponding masks. GenSeg's superior OOD generalization

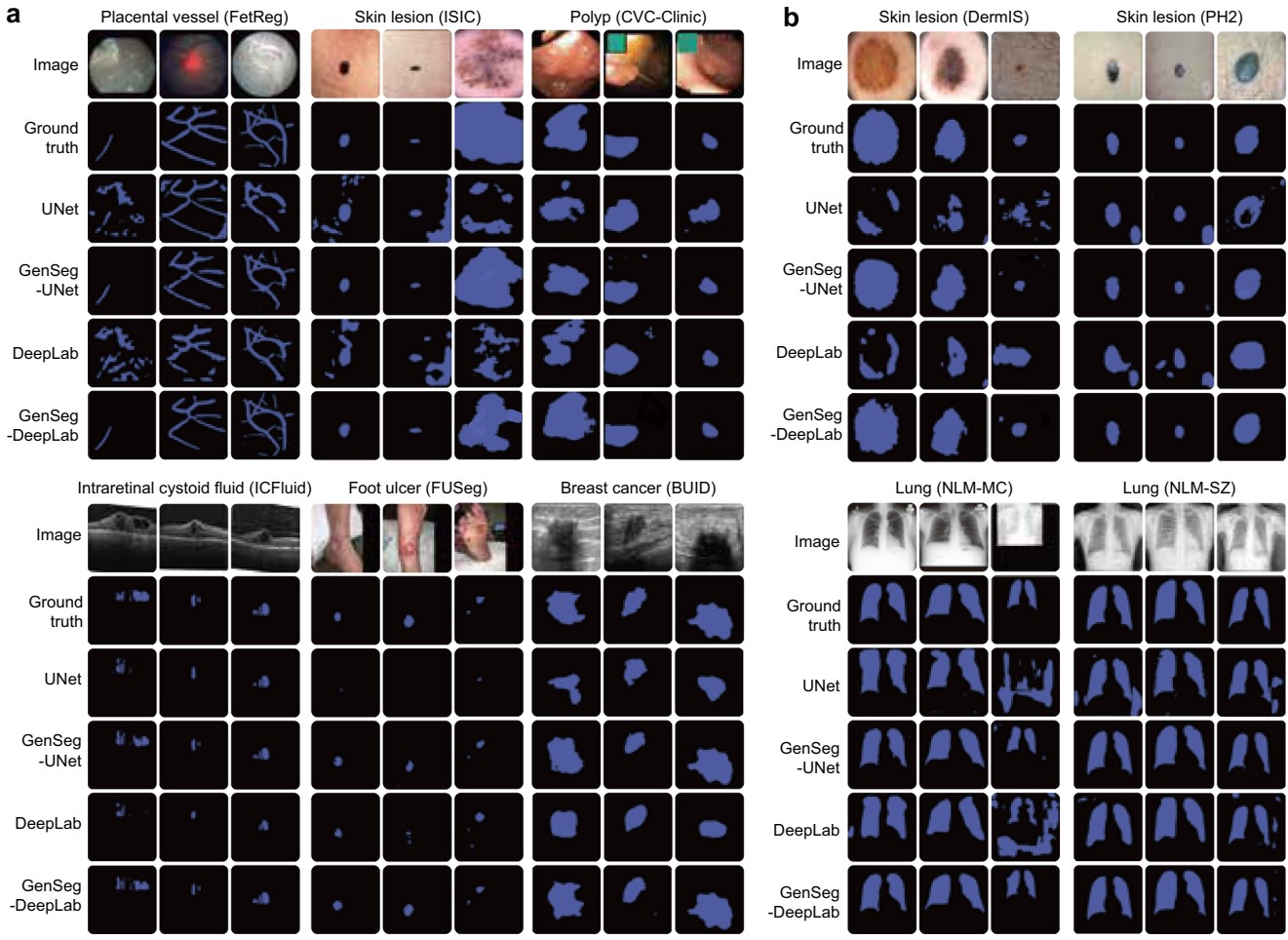

**Fig. 3 | GenSeg improves in-domain and out-of-domain generalization performance across a variety of segmentation tasks covering diverse diseases, organs, and imaging modalities. a** Visualizations of segmentation masks predicted by GenSeg-DeepLab and GenSeg-UNet under in-domain settings in the tasks of segmenting placental vessels, skin lesions, polyps, intraretinal cystoid fluids, foot ulcers, and breast cancer using limited training data (50, 40, 40, 50, 50, and 100 examples from the FetReg, ISIC, CVC-Clinic, ICFluid, FUSeg, and BUID datasets), compared to vanilla UNet and DeepLab. **b** Visualizations of segmentation masks predicted by GenSeg-DeepLab and GenSeg-UNet under out-of-domain settings in segmenting skin lesions (using only 40 examples from the ISIC dataset for training, and the DermIS and PH2 datasets for testing) and lungs (using only 9 examples from the JSRT dataset for training, and the NLM-MC and NLM-SZ datasets for testing), compared to vanilla UNet and DeepLab. All qualitative examples are sourced from publicly available datasets.

capability stems from its ability to generate diverse medical images accompanied by precise segmentation masks. When trained on this diverse augmented dataset, segmentation models can learn more robust and OOD generalizable feature representations.

## GenSeg achieves comparable performance to baselines with significantly fewer training examples

In comparing the number of training examples required for GenSeg and baseline models to achieve similar performance, GenSeg consistently required fewer examples. Figure 4 illustrates this point by plotting segmentation performance (y-axis) against the number of training examples (x-axis) for various methods. Methods that are closer to the upper left corner of the subfigure are considered more sample-efficient, as they achieve superior segmentation performance with fewer training examples. Across all subfigures, our methods consistently position nearer to these optimal upper left corners compared to the baseline methods. First, GenSeg demonstrates superior sample efficiency under in-domain settings (Fig. 4a). For example, in the placental vessel segmentation task, GenSeg-DeepLab achieved a Dice score of 0.51 with only 50 training examples, a tenfold reduction compared to DeepLab's 500 examples needed to reach the same score. In foot ulcer segmentation, to reach a Dice score around 0.6,

UNet needed 600 examples, in contrast to GenSeg-UNet, which required only 50 examples, a twelve-fold reduction. DeepLab required 800 training examples for a Dice score of 0.73, whereas GenSeg-DeepLab achieved the same score with only 100 examples, an eight-fold reduction. In lung segmentation, achieving a Dice score of 0.97 required 175 examples for UNet, whereas GenSeg-UNet needed just 9 examples, representing a 19-fold reduction. Second, the sample efficiency of GenSeg is also evident in OOD settings (Fig. 4b). For example, in lung segmentation, achieving an OOD generalization performance of 0.93 on the NLM-SZ dataset required 175 training examples from the JSRT dataset for UNet, while GenSeg-UNet needed only 9 examples, representing a 19-fold reduction. In skin lesion segmentation, GenSeg-DeepLab, trained with only 40 ISIC examples, reached a Jaccard index of 0.67 on DermIS, a performance that DeepLab could not match even with 200 examples.

## GenSeg outperforms widely used data augmentation and generation tools

We compared GenSeg against prevalent data augmentation methods, including rotation, flipping, and translation, as well as their combinations. Furthermore, GenSeg was benchmarked against a data generation approach[27], which is based on the Wasserstein Generative

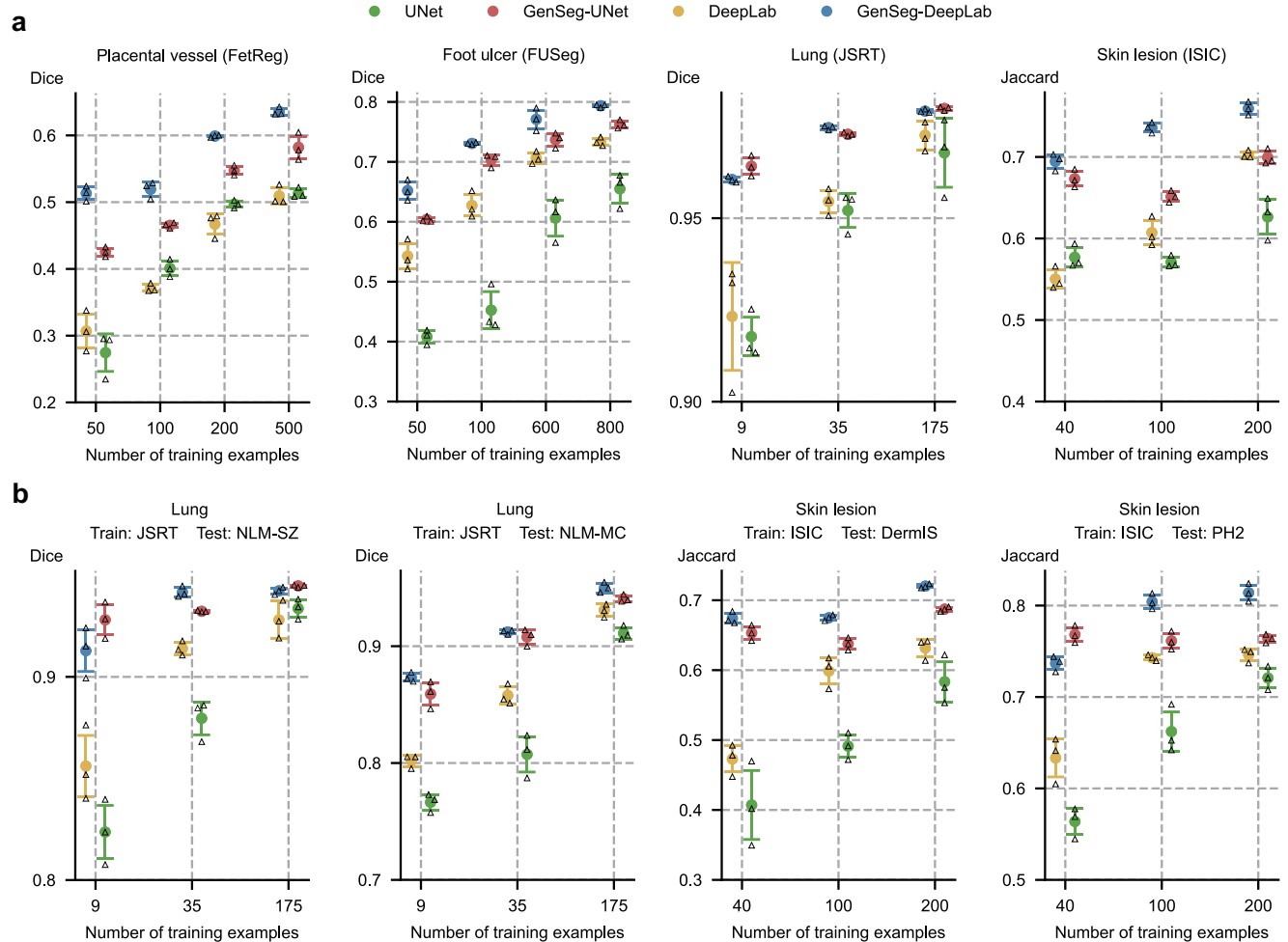

**Fig. 4 | GenSeg achieves performance on par with baseline models while requiring significantly fewer training examples. a** The in-domain generalization performance of GenSeg-UNet and GenSeg-DeepLab with different numbers of training examples from the FetReg, FUSeg, JSRT, and ISIC datasets in segmenting placental vessels, foot ulcers, lungs, and skin lesions, compared to UNet and DeepLab. **b** The out-of-domain generalization performance of GenSeg-UNet and GenSeg-DeepLab with different numbers of training examples in segmenting lungs

(using examples from JSRT for training, and NLM-SZ and NLM-MC for testing) and skin lesions (using examples from ISIC for training, and DermIS and PH2 for testing), compared to UNet and DeepLab. In all panels, bar heights represent the mean, and error bars indicate the standard deviation across three independent runs with different random seeds. Results from individual runs are shown as black triangles. Source data are provided as a Source Data file.

Adversarial Network (WGAN)[28]. For each baseline augmentation method, the same hyperparameters (e.g., rotation angle) were consistently applied to both the input image and the corresponding output mask within each training example, resulting in augmented image-mask pairs. GenSeg significantly surpassed these methods under in-domain settings (Fig. 5a and Supplementary Fig. 2). For instance, in foot ulcer segmentation using UNet as the backbone segmentation model, GenSeg attained a Dice score of 0.74, significantly surpassing the top baseline method, WGAN, which achieved 0.66. Similarly, in polyp segmentation with DeepLab, GenSeg scored 0.76, significantly outperforming the best baselines—Flip, Combine, and WGAN—which scored 0.69. GenSeg also demonstrated superior OOD generalization performance compared to the baselines (Fig. 5c and Supplementary Fig. 3b). For instance, in UNet-based skin lesion segmentation, with 40 training examples from the ISIC dataset, GenSeg achieved a Dice score of 0.77 on the PH2 dataset, substantially surpassing the best-performing baseline, Flip, which scored 0.68. Moreover, GenSeg demonstrated comparable performance to baseline methods with fewer training examples (Fig. 5b and Supplementary Fig. 3a) under in-domain settings. For instance, using only 40 training examples for skin lesion segmentation with UNet, GenSeg achieved a Dice score of 0.67. In contrast, the best performing baseline, Combine, required 200

examples to reach the same score. Similarly, with fewer training examples, GenSeg achieved comparable performance to baseline methods under OOD settings (Fig. 5c and Supplementary Fig. 3b). For example, in lung segmentation with UNet, GenSeg reached a Dice score of 0.93 using just 9 training examples, whereas the best performing baseline required 175 examples to achieve a similar score.

GenSeg outperforms existing data augmentation and generation techniques primarily due to its end-to-end data generation mechanism. Unlike previous methods that separate data augmentation/generation from segmentation model training, our approach integrates them end-to-end within a unified, MLO framework. Within this framework, the validation performance of the segmentation model acts as a direct indicator of the generated data's usefulness. By leveraging this performance to inform the training process of the generation model, we ensure that the data produced is specifically optimized to improve the segmentation model. In previous methods, segmentation performance does not impact the process of data augmentation and generation. As a result, the augmented/generated data might not be effectively tailored for training the segmentation model. Furthermore, our framework learns a generative model that excels in generating data with greater diversity compared to existing augmentation methods.

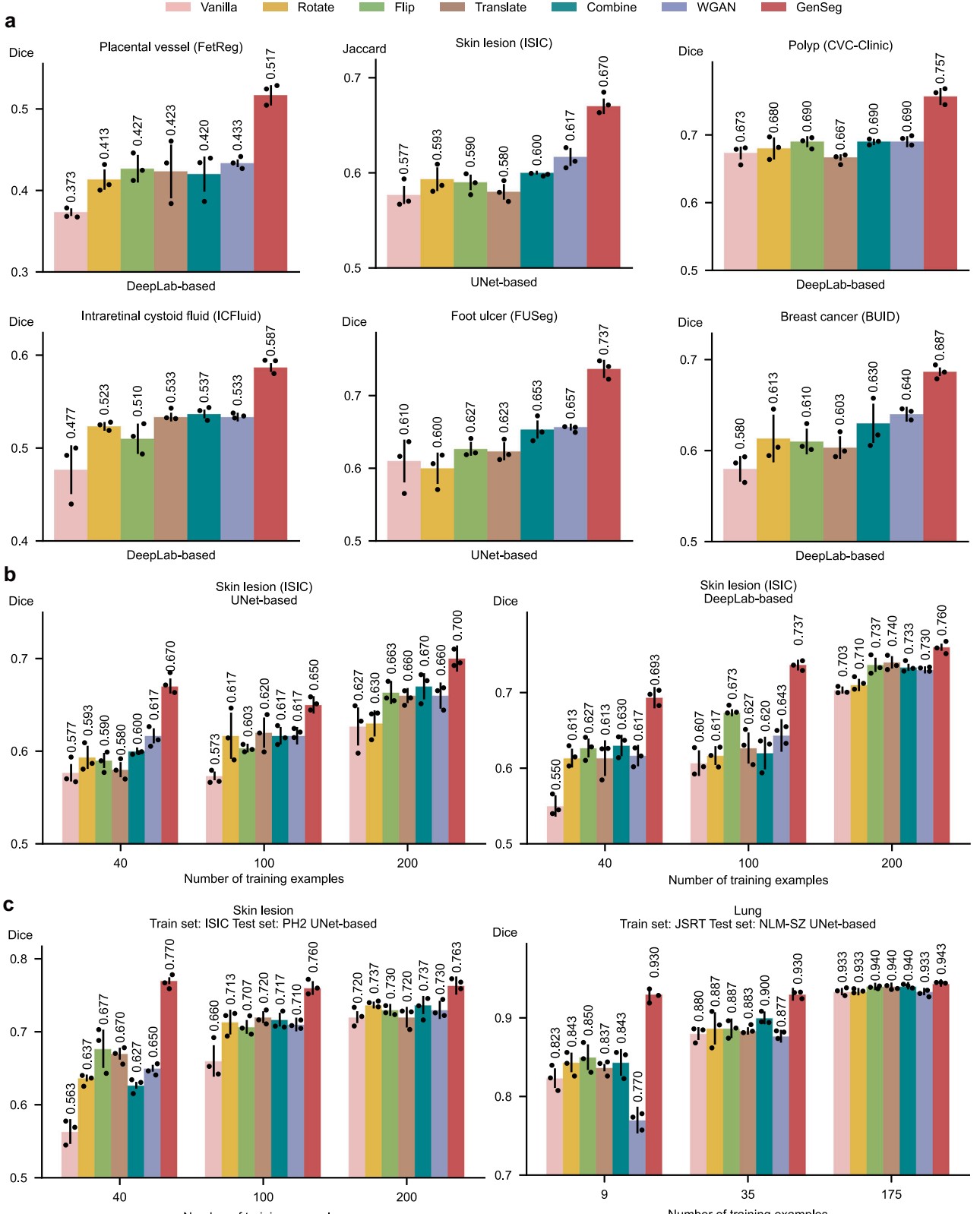

## GenSeg outperforms state-of-the-art semi-supervised segmentation methods

We conducted a comparative analysis of GenSeg against leading semi-supervised segmentation methods[18–20,29], including cross-teaching between convolutional neural networks and Transformer (CTBCT)[30], deep co-training (DCT)[29], and a mutual correction framework (MCF)[31], which employ external unlabeled images (1000 in each experiment) to enhance model training and thereby improve segmentation performance. GenSeg, which does not require any additional unlabeled images, significantly outperformed baseline methods under in-domain settings (Fig. 6a and Supplementary Fig. 4). For example, when using DeepLab as the backbone segmentation model for polyp segmentation, GenSeg achieved a Dice score of 0.76, markedly outperforming the top baseline method, MCF, which reached only 0.69. GenSeg also

**Fig. 5 | GenSeg significantly outperformed widely used data augmentation and generation methods. a** GenSeg's in-domain generalization performance compared to baseline methods, including Vanilla (without any data augmentations), Rotate, Flip, Translate, Combine, and WGAN, when used with UNet or DeepLab in segmenting placental vessels, skin lesions, polyps, intraretinal cystoid fluids, foot ulcers, and breast cancer using the FetReg, ISIC, CVC-Clinic, ICFluid, FUSeg, and BUID datasets. **b** GenSeg's in-domain generalization performance compared to baseline methods using a varying number of training examples from the ISIC dataset for segmenting skin lesions, with UNet and DeepLab as the backbone segmentation models. **c** GenSeg's out-of-domain generalization performance compared to baseline methods across varying numbers of training examples in segmenting lungs (using examples from JSRT for training, and NLM-SZ and NLM-MC for testing) and skin lesions (using examples from ISIC for training, and DermIS and PH2 for testing), with UNet and DeepLab as the backbone segmentation models. In all panels, bar heights represent the mean, and error bars indicate the standard deviation across three independent runs with different random seeds. Results from individual runs are shown as dot points. Source data are provided as a Source Data file.

exhibited superior OOD generalization capabilities compared to baseline methods (Fig. 6c and Supplementary Fig. 5b). For instance, in skin lesion segmentation based on DeepLab with 40 training examples from the ISIC dataset, GenSeg achieved a Dice score of 0.67 on the DermIS dataset, significantly higher than the best-performing baseline, MCF, which scored 0.58. Additionally, GenSeg showed performance on par with baseline methods using fewer training examples in both in-domain (Fig. 6b and Supplementary Fig. 5a) and OOD settings (Fig. 6c and Supplementary Fig. 5b).

In the context of medical imaging, collecting even unlabeled images presents a considerable challenge due to stringent privacy concerns and regulatory constraints (e.g., IRB approval), thereby reducing the feasibility of semi-supervised methods. Despite the use of unlabeled real images, semi-supervised approaches underperform compared to GenSeg. This is primarily because these methods struggle to generate accurate masks for unlabeled images, meaning that they are less effective at creating labeled training data. In contrast, GenSeg is capable of producing high-quality images from masks, ensuring a close correspondence between the images' contents and the masks, thereby efficiently generating labeled training examples.

## GenSeg's end-to-end generation mechanism is superior to baselines' separate generation

We compared the effectiveness of GenSeg's end-to-end data generation mechanism against a baseline approach, Separate, which separates data generation from segmentation model training. In Separate, the mask-to-image generation model is initially trained and then fixed. Subsequently, it generates data, which is then utilized to train the segmentation model. The end-to-end GenSeg framework consistently outperformed the Separate approach under both in-domain (Fig. 7a and Supplementary Fig. 6a) and OOD settings (Fig. 7b and Supplementary Fig. 6b). For instance, in the segmentation of placental vessels, GenSeg-DeepLab attained an in-domain Dice score of 0.52, significantly surpassing Separate-DeepLab, which scored 0.42. In lung segmentation using JSRT as the training dataset, GenSeg-UNet achieved an OOD Dice score of 0.93 on the NLM-SZ dataset, considerably better than the 0.84 scored by Separate-UNet.

## GenSeg outperforms nnUNet across both in-domain and out-of-domain scenarios

We compared GenSeg-UNet with nnUNet[2] - a state-of-the-art method for medical image segmentation - under both in-domain and OOD settings across multiple segmentation tasks. GenSeg-UNet consistently outperformed nnUNet in these data-scarce scenarios (Fig. 8a, b). In in-domain scenarios (Fig. 8a), GenSeg-UNet achieves 1–7% (absolute percentages) higher performance scores across all tasks. In OOD evaluations (Fig. 8b), which involve more substantial distributional shifts, GenSeg-UNet demonstrates even greater improvements across all tasks, outperforming nnUNet by 5–16% (absolute percentages). For instance, in the lung segmentation task, when trained on only 175 examples from the JSRT dataset and evaluated on the SZ dataset, GenSeg-UNet achieves a Dice score of 94.5%, compared to 78.4% with nnUNet—a substantial gain of 16.1%.

The superior performance of GenSeg over nnUNet in ultra-low data regimes can be attributed to fundamental differences in their

augmentation strategies. nnUNet employs standard augmentation techniques such as rotation, scaling, Gaussian blur, and intensity adjustments, which, while effective in moderate- to large-scale data settings, offer limited diversity and adaptability in severely data-constrained scenarios. In contrast, GenSeg trains a deep generative model that synthesizes diverse and semantically consistent image-mask pairs tailored to the specific task and dataset. This generative augmentation approach introduces significantly greater variability into the training data, enabling the segmentation model to learn more robust and generalizable representations. By aligning the data generation process with segmentation performance through end-to-end MLO, GenSeg ensures that the synthesized data is not only realistic but also highly informative for improving downstream segmentation accuracy.

## GenSeg improves the performance of diverse backbone segmentation models

GenSeg is a versatile, model-agnostic framework that can seamlessly integrate with segmentation models with diverse architectures to improve their performance. For example, after applying our framework on UNet and DeepLab, we observed significant enhancements in their performance (Figs. 2–7), both for in-domain and OOD settings. Furthermore, we also integrated this framework with a Transformer-based segmentation model, SwinUnet[32]. Using just 40 training examples from the ISIC dataset, GenSeg-SwinUnet achieved a Jaccard index of 0.62 on the ISIC test set. Furthermore, it demonstrated strong generalization with OOD Jaccard index scores of 0.65 on the PH2 dataset and 0.62 on the DermIS dataset. These results represent a substantial improvement over the baseline SwinUnet model, which achieved Jaccard indices of 0.55 on ISIC, 0.56 on PH2, and 0.38 on DermIS (Fig. 8c).

## GenSeg improves 3D medical image segmentation

In addition to 2D medical image segmentation, GenSeg can be extended to support 3D segmentation tasks. To enable this, we adapted our framework by incorporating 3D UNet[33] as the segmentation model and Pix2PixNIfTI[34] as the generative model, facilitating joint generation and segmentation in a 3D volumetric setting. We make the architecture of the Pix2PixNIfTI model searchable by replacing the convolution and transposed convolution layers in the original generator with our differentiable convolutional and transposed convolutional cells. The architecture parameters of the modified Pix2PixNIfTI model are optimized by minimizing the segmentation loss on the validation set within our MLO-based framework. During training, the input 3D masks are first augmented using rotation and flipping transformations, and the generator then synthesizes 3D volumes from these augmented masks. We evaluated this 3D extension on two datasets from the Medical Segmentation Decathlon (MSD) challenge[4], focusing on hippocampus and liver segmentation tasks. Experiments were conducted under both ultra-low data settings (40 training volumes) and higher data settings using the full available training sets (208 volumes for hippocampus and 98 for liver).

GenSeg consistently improved segmentation performance over the baseline 3D UNet in both regimes (Fig. 8d). Notably, in the ultra-low data setting, GenSeg yielded substantial gains, demonstrating its robustness and effectiveness in data-constrained 3D segmentation

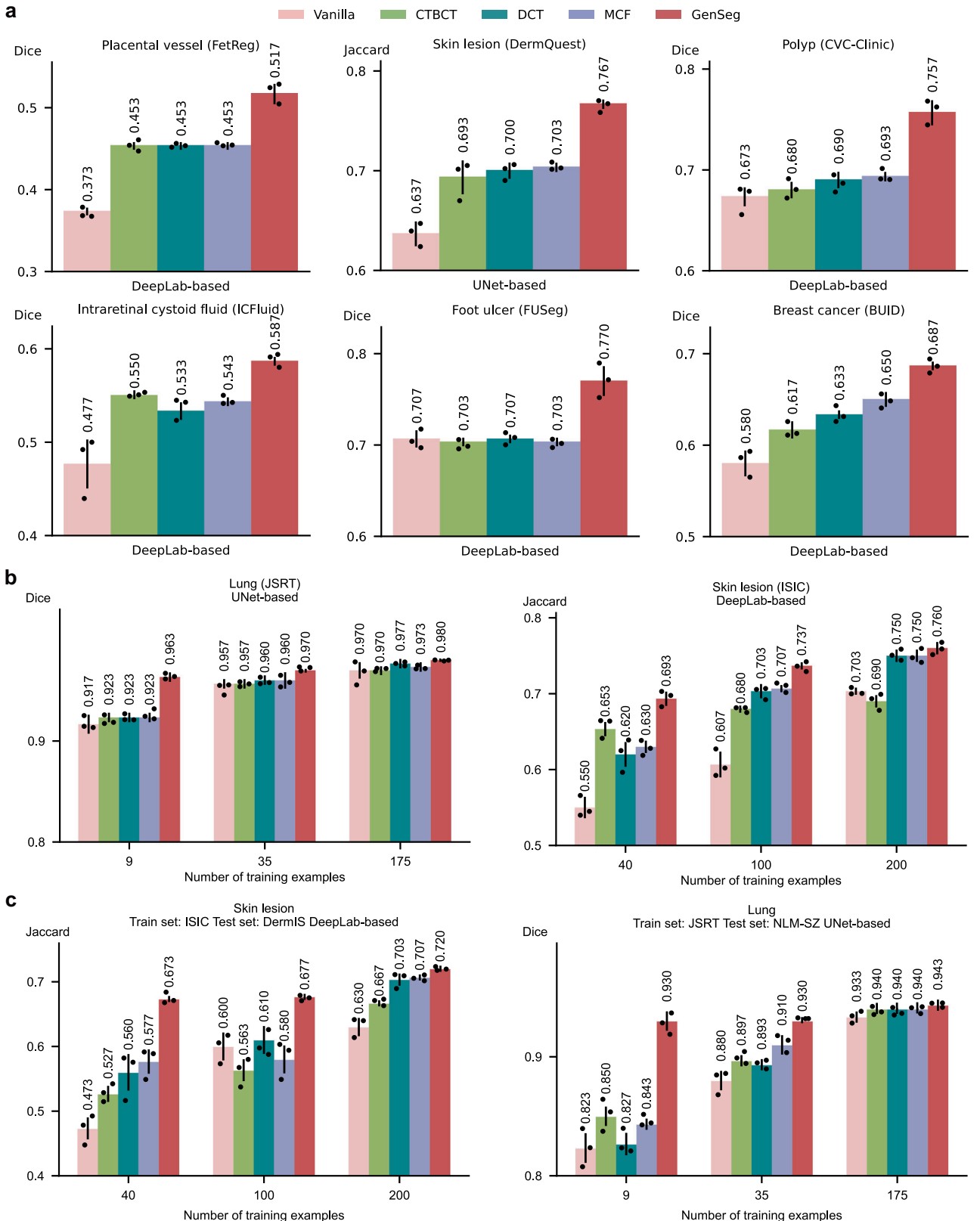

tasks. These results confirm that GenSeg generalizes beyond 2D segmentation and remains effective when applied to more complex 3D volumetric data.

## GenSeg is effective in high-data regimes as well

While GenSeg is designed to enable medical image segmentation in ultra-low data regimes, we further investigated its effectiveness in higher data settings. We conducted experiments on the ISIC, FetReg, BUID, and CVC-Clinic datasets using UNet as the segmentation model. Two training regimes were evaluated: (1) UNet-low and GenSeg-UNet-low, trained under ultra-low data conditions with 40, 50, 100, and 40 training examples from the respective datasets; and (2) UNet-high and GenSeg-UNet-high, trained using the full available training sets, consisting of 1000, 2000, 400, and 400 examples, respectively.

**Fig. 6 | GenSeg significantly outperformed state-of-the-art semi-supervised segmentation methods. a** GenSeg's in-domain generalization performance compared to baseline methods, including Vanilla (UNet/DeepLab), CTBCT, DCT, and MCF, when used with UNet or DeepLab in segmenting placental vessels, skin lesions, polyps, intraretinal cystoid fluids, foot ulcers, and breast cancer utilizing the FetReg, DermQuest, CVC-Clinic, ICFluid, FUSeg, and BUID datasets. **b** GenSeg's in-domain generalization performance compared to baseline methods using a varying number of training examples from the ISIC and JSRT datasets for segmenting skin lesions and lungs, with UNet and DeepLab as the backbone segmentation models. **c** GenSeg's out-of-domain generalization performance compared to baseline methods across varying numbers of training examples in segmenting lungs (using examples from JSRT for training, and NLM-SZ and NLM-MC for testing) and skin lesions (using examples from ISIC for training, and DermIS and PH2 for testing), with UNet and DeepLab as the backbone segmentation models. In all panels, bar heights represent the mean, and error bars indicate the standard deviation across three independent runs with different random seeds. Results from individual runs are shown as dot points. Source data are provided as a Source Data file.

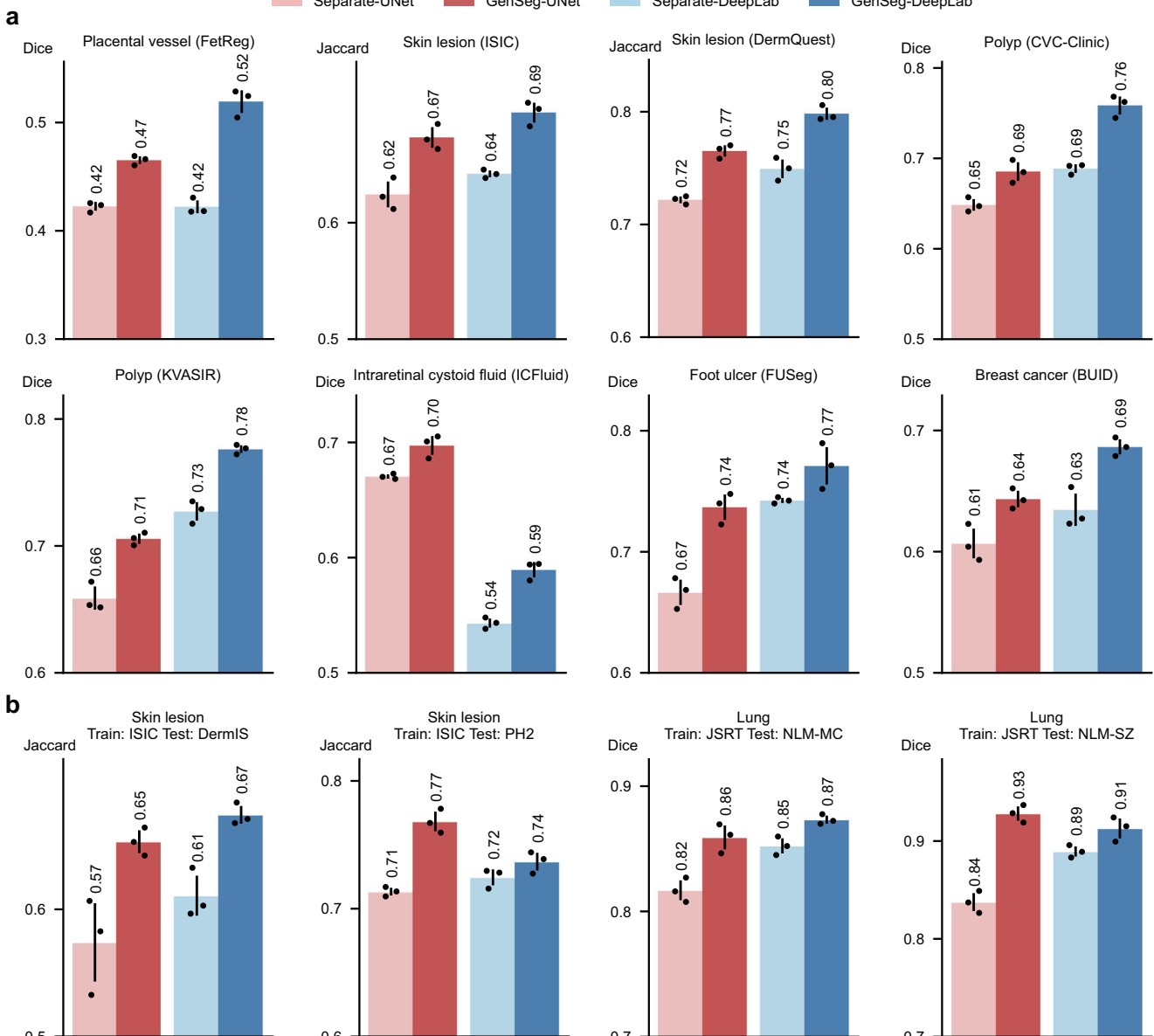

**Fig. 7 | GenSeg's end-to-end data generation mechanism significantly outperformed baselines' separate generation mechanism. a** The in-domain generalization performance of GenSeg, which performs data generation and segmentation model training end-to-end, compared to the Separate baseline, which performs the two processes separately, when used with UNet or DeepLab in segmenting placental vessels, skin lesions, polyps, intraretinal cystoid fluids, foot ulcers, and breast cancer utilizing the FetReg, ISIC, DermQuest, CVC-Clinic, KVASIR, ICFluid, FUSeg, and BUID datasets. **b** GenSeg's out-of-domain generalization performance compared to the Separate baseline in segmenting skin lesions (using examples from ISIC for training, and DermIS and PH2 for testing) and lungs (using examples from JSRT for training, and NLM-SZ and NLM-MC for testing), with UNet and DeepLab as the backbone segmentation models. In all panels, bar heights represent the mean, and error bars indicate the standard deviation across three independent runs with different random seeds. Results from individual runs are shown as dot points. Source data are provided as a Source Data file.

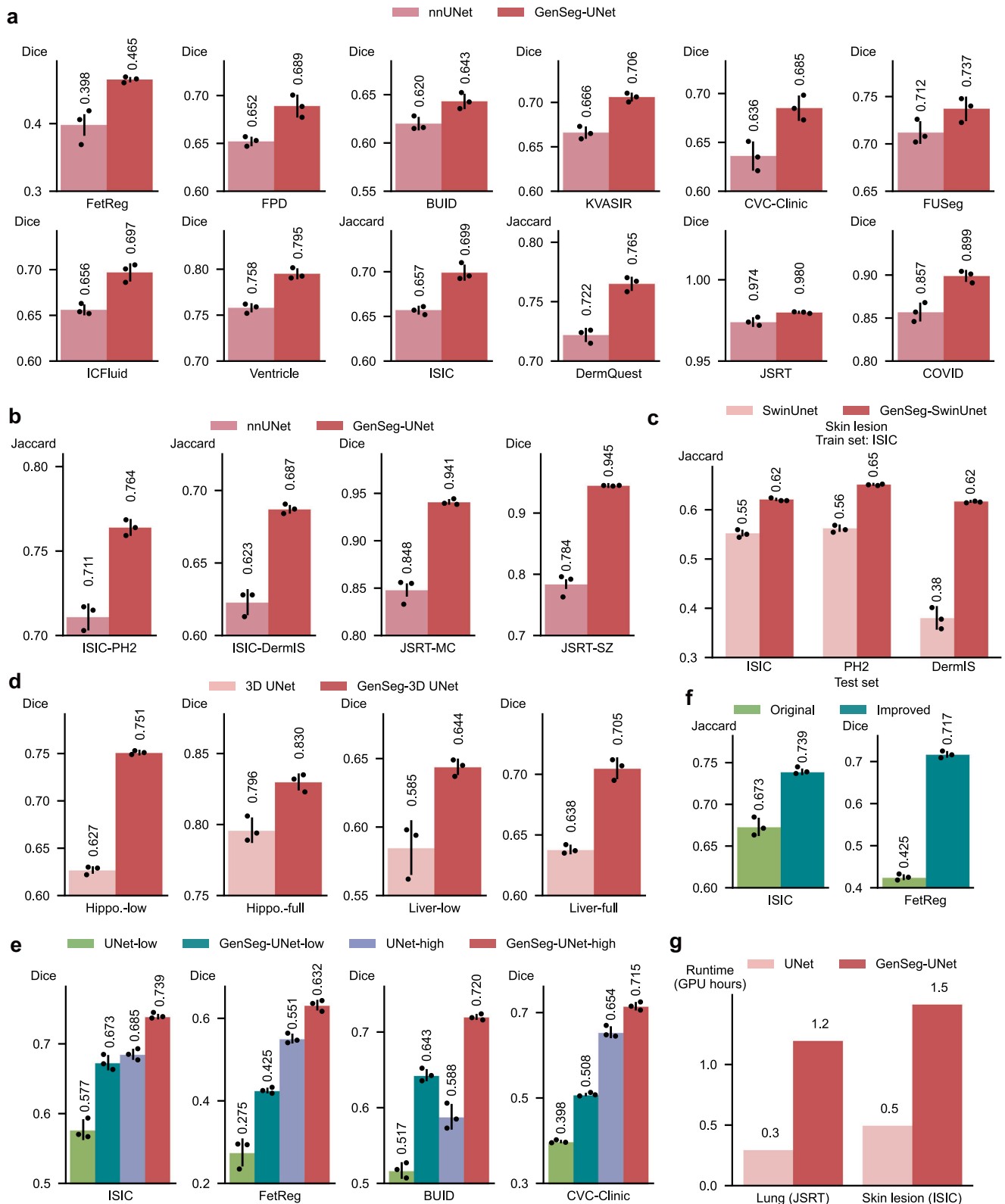

As shown in Fig. 8e, several key observations emerge. First, GenSeg-UNet-high outperforms UNet-high across all datasets, demonstrating that GenSeg's generative augmentation strategy continues to provide benefits even in high-data regimes. Second, as expected, segmentation performance improves for all models as the training set size increases. Third, despite being trained on significantly fewer examples, GenSeg-UNet-low achieves performance that is often close to that of UNet-high, highlighting GenSeg's

strength in data-scarce scenarios. These findings underscore the versatility and effectiveness of the GenSeg framework across varying data availability conditions. GenSeg consistently enhances segmentation performance regardless of dataset size by integrating generative augmentation into an end-to-end, task-driven learning paradigm. While particularly valuable in low-data regimes, GenSeg also improves generalization in more data-rich settings by enriching the training signal.

**Fig. 8 | GenSeg consistently enhances segmentation performance across diverse tasks, domains, and data regimes. a** GenSeg-UNet consistently outperforms nnUNet across a range of segmentation tasks under in-domain scenarios. **b** GenSeg-UNet consistently demonstrates superior performance to nnUNet across diverse segmentation tasks in out-of-domain settings. In the X-Y notation, *X* refers to the training dataset and *Y* to the test dataset, where *X* and *Y* are from distinct distributions. **c** GenSeg-SwinUnet outperforms SwinUnet, both trained on 40 examples from the ISIC dataset and evaluated on the test sets of ISIC, PH2, and DermIS. **d** Extension of the GenSeg framework to 3D medical image segmentation tasks under different training data regimes. "Hippo.-low" refers to training with an ultra-low data setting for hippocampus segmentation, while "Hippo.-full" refers to training with the full available dataset. The same settings are applied to the liver segmentation task. **e** Comparison of model performance under ultra-low and high data regimes. "UNet-low" denotes the UNet model trained with an ultra-low amount of data, while "UNet-high" refers to the model trained with the full available dataset. The same training settings are applied to GenSeg-UNet. **f** GenSeg's performance on the ISIC and FetReg datasets can be further improved by employing several strategies, including increasing the number of training examples, using task-appropriate segmentation models, and refining augmentation techniques. **g** The runtime (in hours on an A100 GPU) of GenSeg-UNet was measured for lung segmentation using JSRT as the training data and for skin lesion segmentation using ISIC as the training data. In all panels (except **g**), bar heights represent the mean, and error bars indicate the standard deviation across three independent runs with different random seeds. Results from individual runs are shown as dot points. Source data are provided as a Source Data file.

## Further improvement on ISIC and FetReg datasets

To further enhance GenSeg's segmentation performance on challenging datasets such as ISIC and FetReg, we conducted additional experiments by incorporating several targeted strategies. These included increasing the amount of training data, refining augmentation techniques, and employing a more proper segmentation backbone. For the ISIC dataset (UNet was used as the segmentation model), we increased the number of training examples from 40 to 1000, which led to an improvement in Jaccard score from 67.3% to 73.9% (Fig. 8f), reaching a level considered satisfactory for binary segmentation tasks. For the FetReg dataset, which presents unique challenges due to high anatomical variability, low image contrast, and the complexity of placental vessel structures, we implemented three modifications: narrowing the rotation augmentation range to (−5° to 5°), replacing UNet with DeepLab as the segmentation model, and expanding the training set size from 50 to 2000 examples. These adjustments resulted in a significant performance gain, improving the Dice score to 71.7% (Fig. 8f). These findings indicate that with sufficient data and appropriate architectural and augmentation refinements, GenSeg can achieve high segmentation accuracy even in complex tasks.

## Ablation study evaluating different mask-to-image generative models

We conducted ablation studies to investigate how different choices of mask-to-image generative models affect the final segmentation performance. In addition to the GAN-based Pix2Pix model used in our current framework, we evaluated two state-of-the-art alternatives: Soft-Intro VAE[35], a variational autoencoder (VAE)[36–39] based model, and BBDM[40], a diffusion-based generative model[41]. We integrated each model into our GenSeg framework by using them to replace the original Pix2Pix mask-to-image generator. We modified both BBDM and Soft-Intro VAE by incorporating our multi-branch convolutional cells into their generator networks, to allow their architectures to be optimized based on segmentation performance. We trained each model using two strategies: (1) Separate, where the generative model is trained independently and then fixed before segmentation model training, and (2) End2End, our proposed MLO framework. Evaluation was performed under both in-domain and OOD scenarios.

BBDM (End2End) achieved the highest performance across all datasets, under both in-domain settings (Fig. 9a) and OOD settings (Fig. 9b). The performance of Pix2Pix (End2End) and Soft-Intro VAE (End2End) was comparable, with both trailing slightly behind BBDM. However, BBDM incurs significantly higher computational cost and model size compared to both Pix2Pix and Soft-Intro VAE under the End2End strategy (Fig. 9c). Considering the trade-off between segmentation performance and computational efficiency, Pix2Pix remains a practical and effective choice for our setting, particularly when computational resources are limited. Furthermore, all three End2End approaches consistently outperformed their respective Separate counterparts, highlighting the advantage of jointly optimizing the generative and segmentation models within an end-to-end training framework. This result reinforces the central premise of GenSeg: that aligning the data generation process with downstream segmentation performance leads to more effective learning.

In addition, within the GAN family, we compared the Pix2Pix model with two other GAN-based models: SPADE[42] and ASAPNet[43]. For a fair comparison, we also made the generator architectures of these models searchable by applying the multi-branch convolutional modification (Fig. 1c) to their generators. Pix2Pix and SPADE demonstrated comparable performance, both significantly outperforming ASAPNet (Fig. 9d). This performance gap can be attributed to the superior image generation capabilities of Pix2Pix and SPADE.

## Ablation study investigating the impact of generating images and masks jointly

In our current framework, image and mask generation is performed using a two-step approach: we first generate augmented masks from real masks using standard augmentation techniques, and then synthesize images from the augmented masks using a mask-to-image generative model. As an alternative, one can generate both the image and the corresponding mask simultaneously[44]. To investigate which strategy is more effective, we compared our two-step approach with an ablation setting referred to as Simultaneous, in which images and masks are generated jointly using the WGAN-GP model[28], integrated within our framework when using UNet as the segmentation model. In this setting, WGAN-GP takes a random noise vector sampled from a Gaussian distribution as input and simultaneously produces a medical image and its corresponding mask. To maintain architectural consistency with our framework, we modified the original WGAN-GP by replacing its convolutional layers with our multi-branch convolutional cells. We then trained the model using our end-to-end optimization strategy to ensure a fair comparison.

The two-step approach consistently outperforms the WGAN-GP-based simultaneous generation method in both in-domain (Fig. 9e) and OOD (Fig. 9f) settings. Notably, in the OOD evaluations—where 40 examples from the ISIC dataset were used for training and PH2 and DermIS served as test sets—the two-step method achieved 12.1% and 8.9% higher performance, respectively.

The superior performance of the two-step approach over the simultaneous generation method can be attributed to the explicit conditioning and structural alignment enforced during the data generation process. In the two-step pipeline, segmentation masks are first augmented and then used as conditioning inputs to guide the image generation process. This explicit conditioning enables the mask-to-image generation model to synthesize images that are tightly aligned with the structural boundaries and semantics defined by the input mask. As a result, the generated image-mask pairs exhibit high spatial coherence and fidelity, which is crucial for effective segmentation model training. In contrast, the simultaneous generation approach, as implemented with WGAN-GP, synthesizes both the image and the mask jointly without enforcing a strong pixel-wise correspondence between the two outputs. This lack of explicit conditioning can lead to

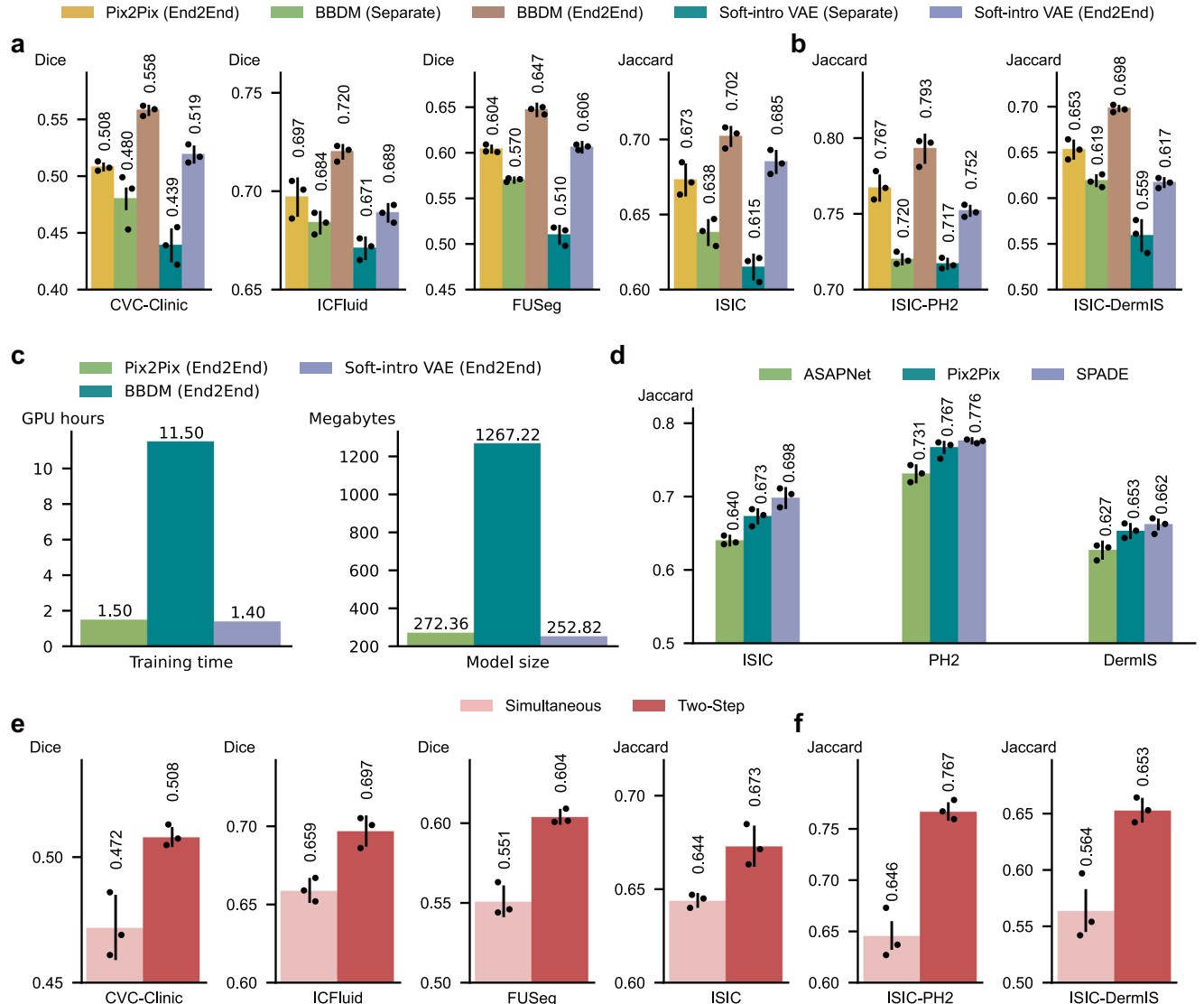

**Fig. 9 | Ablation studies on generative models and generation strategies in GenSeg. a, b** Ablation study evaluating the effectiveness of different generative models - including Pix2Pix (GAN-based), BBDM (diffusion-based), and Soft-Intro VAE (VAE-based) - under separate and end-to-end training strategies. Evaluations were conducted under both in-domain (**a**) and out-of-domain (**b**) scenarios, using UNet as the segmentation model. For out-of-domain scenarios, datasets are labeled in the format X-Y, where X denotes the training dataset and Y denotes the test dataset. **c** Comparison of training time (left) measured on an A100 GPU and model size (right) for Pix2Pix, BBDM, and Soft-Intro VAE within our end-to-end training framework, in skin lesion segmentation with 40 training examples from the ISIC dataset when using UNet as the segmentation model. **d** Impact of mask-to-image GAN models on the performance of GenSeg-UNet was evaluated on the test datasets of ISIC, PH2, and DermIS, in skin lesion segmentation. GenSeg-UNet was trained using 40 examples from the ISIC training dataset. **e, f** Ablation study comparing simultaneous image-mask generation with the two-step approach, where masks are first augmented and then used to generate images. The two-step strategy outperforms simultaneous generation. Experiments were conducted under both in-domain (**e**) and out-of-domain (**f**) settings. In all panels (except **c**), bar heights represent the mean, and error bars indicate the standard deviation across three independent runs with different random seeds. Results from individual runs are shown as dot points. Source data are provided as a Source Data file.

weaker structural alignment, especially in low-data regimes where the model may struggle to learn accurate joint representations. Specifically, it does not impose semantic constraints that guarantee the generated masks accurately delineate regions of interest within the corresponding images. This misalignment can reduce the effectiveness of the generated data in training downstream segmentation models.

## The impact of mask augmentation operations on segmentation performance

In GenSeg, the initial step involves applying augmentation operations to generate synthetic segmentation masks from real masks. We explored the impact of augmentation operations on segmentation performance.

GenSeg, which utilizes all three operations—rotation, translation, and flipping—is compared against three specific ablation settings where only one operation (Rotate, Translate, or Flip) is used to augment the masks. GenSeg demonstrated significantly superior performance compared to any of the individual ablation settings (Fig. 10a). Notably, GenSeg exhibited superior generalization on OOD data, highlighting the advantages of integrating multiple augmentation operations compared to using a single operation. By combining various augmentation operations, GenSeg can generate a broader diversity of augmented masks, which in turn produces a more diverse set of augmented images. Training segmentation models on this diverse dataset allows for learning more robust representations, thereby significantly enhancing generalization capabilities on OOD test data.

## Ablation study on elastic and deformable augmentations

Elastic and deformable augmentations have recently shown promise in enhancing medical image segmentation performance[45]. To evaluate their effectiveness within our framework, we conducted an ablation study assessing the impact of incorporating elastic augmentation into the training pipeline when using UNet as the segmentation model. Specifically, we compared the following three ablation settings: 91) Without Elastic, using only our original set of augmentations (e.g., flipping, rotation, translation), (2) With Elastic, combining our original augmentations with elastic augmentation, and 93) Only Elastic, using elastic augmentation alone, without any other augmentations.

The combination of elastic and traditional augmentations (With Elastic) resulted in modest performance improvements across both in-domain (Fig. 10b) and OOD (Fig. 10c) settings. However, the Without Elastic setting—using only our original traditional augmentations—consistently outperformed the Only Elastic setting (Fig. 10b, c), which applies elastic deformation alone, across all tasks. One possible explanation is that elastic augmentation, when used in isolation, may result in a narrower range of transformations, focusing primarily on localized shape distortions. While such deformations can be beneficial in mimicking anatomical variability, they may not capture broader appearance and geometric changes—such as orientation, scale, or intensity shifts—that traditional augmentations introduce. As a result, relying solely on elastic transformations might limit the diversity of the training data and reduce generalization. These results suggest that traditional augmentations provide a strong and versatile baseline, and that combining them with elastic augmentations may offer additional benefits depending on the dataset characteristics and task requirements.

## Ablation study on the impact of rotation augmentation in placental vessel segmentation

In placental vessel segmentation, the orientation of vessels is highly sensitive, raising concerns that rotation-based augmentations may be unsuitable for such images. To investigate this, we conducted an ablation study on two vessel segmentation datasets: FetReg and FPD, each using 100 training examples. We tested the impact of different degrees of rotation augmentation by comparing five settings: no rotation, small-angle rotation ($-5°$ to $5°$), moderate rotation ($-15°$ to $15°$), large rotation ($-30°$ to $30°$), and very large rotation ($-45°$ to $45°$).

As shown in Fig. 10d, on the FPD dataset, all degrees of rotation yielded better performance than the no-rotation baseline. On the FetReg dataset, small-angle rotation ($-5°$ to $5°$) provided the best performance, while increasing the rotation range gradually led to performance degradation. These observations indicate that large-angle rotations can distort vessel morphology and interfere with fine-grained structural cues essential for accurate segmentation, particularly in tasks requiring high spatial precision. On the other hand, small-angle rotations appear beneficial. They introduce controlled variability that helps improve model generalization without compromising anatomical integrity. We hypothesize that such mild transformations encourage robustness to minor viewpoint changes while still preserving the spatial structure of vessels—an important consideration in vascular imaging. In summary, our results confirm that vessel segmentation tasks are sensitive to large rotational transformations, which can negatively impact performance. However, mild rotations in the range of $-5°$ to $5°$ strike a balance between augmentation diversity and structural preservation, leading to improved outcomes.

## Ablation study on learnable multi-branch convolutions

To quantify the impact of the multi-branch design in Fig. 1c, we conducted an ablation study involving three settings. In the first setting (Single-branch), we trained a standard single-branch Pix2Pix generator to synthesize images, which were then used to train the segmentation model in a separate stage. In the second setting (Fixed Multi-branch), we used a multi-branch Pix2Pix generator with branch weights (i.e., all weights $\alpha$ in Fig. 1c) fixed to 1, also trained independently from the segmentation model. In the third setting (Learnable Multi-branch), which corresponds to our full GenSeg framework, the generator was integrated into an end-to-end pipeline, where the branch weights $\alpha$ were learned by minimizing segmentation loss on the validation set. We evaluated all three configurations on three representative tasks: skin lesion segmentation (ISIC dataset, 200 training examples), intraretinal cystoid segmentation (ICFluid dataset, 50 training examples), and breast cancer segmentation (BUID dataset, 100 training examples). As shown in Fig. 10e, the Fixed Multi-branch model consistently outperformed the Single-branch model, demonstrating the advantage of using multi-branch convolutions. Moreover, the Learnable Multi-branch model further improved performance, highlighting the benefit of learning the branch weights in a task-adaptive manner. To assess the statistical significance of these improvements, we conducted two-sided paired $t$-tests on performance scores across three tasks. As shown in Supplementary Table 2, each method was evaluated over three independent training runs with different random seeds, and pairwise comparisons were performed. Most $p$-values are below 0.05, indicating that the performance gains from the multi-branch architecture—particularly the learnable variant—are statistically significant.

We attribute these improvements to the increased representational capacity of the multi-branch architecture, which enables the generator to learn a more diverse set of features tailored to varying spatial and structural characteristics across datasets. While the fixed multi-branch design provides architectural flexibility, the learnable version further strengthens performance by enabling end-to-end optimization that aligns synthetic data generation with the segmentation objective. In summary, this ablation study demonstrates that learnable multi-branch convolutions significantly improve segmentation accuracy, demonstrating their role as an important micro-architectural component of the GenSeg framework.

## The impact of the tradeoff parameter $\gamma$ on segmentation performance

We investigated the effect of the hyperparameter $\gamma$ in Eq. (2) on the performance of our method. This parameter controls the balance between the contributions of real and generated data during the training of the segmentation model. Optimal performance was observed with a moderate $\gamma$ value (e.g., 1), which effectively balanced the use of real and generated data (Fig. 10f).

## Computation costs

Given that GenSeg is designed for scenarios with limited training data, the overall training time is minimal, often requiring less than 2 GPU hours (Fig. 8g). To enhance the efficiency of GenSeg's training, we plan to incorporate strategies from refs. 46,47 for accelerated GAN training and implement the algorithm proposed in ref. 48 to expedite the convergence of MLO. Importantly, our method does not increase the inference cost of the segmentation model. This is because our approach maintains the original architecture of the segmentation model, ensuring that the Multiply-Accumulate (MAC) operations remain unchanged.

## Discussion

We present GenSeg, a robust data generation tool designed for generating high-quality data to enhance the training of medical image segmentation models. Demonstrating superior in-domain and OOD generalization performance across nine diverse segmentation tasks and 19 datasets, GenSeg excels particularly in scenarios with a limited number of real, expert-annotated training examples (as few as 50). GenSeg substantially enhances sample efficiency, requiring far fewer expert-annotated training examples than baseline methods to achieve similar performance. This greatly reduces both the burden and costs associated with medical image annotation.

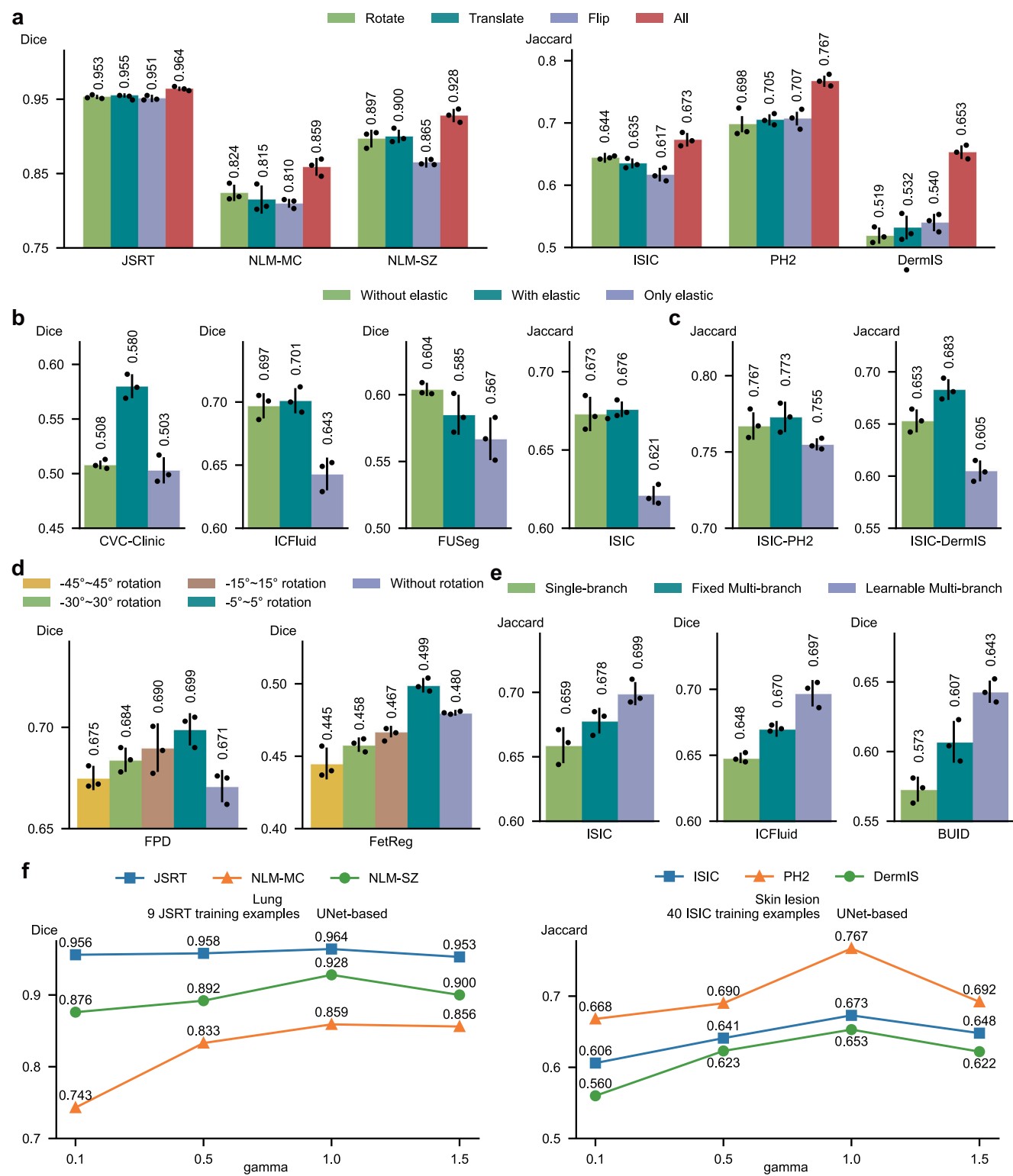

GenSeg stands out by requiring fewer expert-annotated real training examples compared to baseline methods, yet it achieves comparable performance. This substantial reduction in the need for manually labeled segmentation masks significantly cuts down both the burden and costs associated with medical image annotation. With just a small set of real examples, GenSeg effectively trains a data generation model which then produces additional synthetic data, effectively mimicking the benefits of using a large dataset of real examples.

GenSeg significantly improves segmentation models' OOD generalization capability. GenSeg is capable of generating diverse medical images accompanied by precise segmentation masks. When trained on this diverse augmented dataset, segmentation models can learn more robust and OOD generalizable feature representations.

GenSeg stands out from current data augmentation and generation techniques by offering superior segmentation performance, primarily due to its end-to-end data generation mechanism. Unlike previous methods that separate data augmentation/generation and segmentation model training, our approach integrates them end-to-end within a unified, MLO framework. Within this framework, the validation performance of the segmentation model acts as a direct

**Fig. 10 | Ablation studies of augmentation strategies, architectural components, and parameter sensitivity in GenSeg. a** (Left) Impact of augmentation operations on the performance of GenSeg-UNet was evaluated on the test datasets of JSRT, NLM-MC, and NLM-SZ, in lung segmentation. GenSeg-UNet was trained using 9 examples from the JSRT training dataset. ALL refers to the full GenSeg method that incorporates all three operations. (Right) Impact of augmentation operations on the performance of GenSeg-UNet was evaluated on the test datasets of ISIC, PH2, and DermIS, in skin lesion segmentation. GenSeg-UNet was trained using 40 examples from the ISIC training dataset. **b, c** Ablation study evaluating the impact of elastic augmentation under in-domain (**b**) and out-of-domain settings (**c**). In out-of-domain scenarios, datasets are denoted in the format X-Y, where X represents the training dataset and Y the test dataset. UNet was used as the segmentation model. **d** Ablation study evaluating the impact of rotation augmentation on placental vessel segmentation using the FetReg and FPD datasets with UNet as the segmentation model. **e** Ablation study on learnable multi-branch convolutions, with UNet as the segmentation model. **f** (Left) Impact of the tradeoff parameter $\gamma$ on the performance of GenSeg-UNet on the test datasets of JSRT, NLM-MC, and NLM-SZ, in lung segmentation with 9 examples from the JSRT training dataset. (Right) Impact of the tradeoff parameter $\gamma$ on the performance of GenSeg-UNet on the test datasets of ISIC, PH2, and DermIS, in skin lesion segmentation with 40 examples from the ISIC training dataset. In all panels (except **f**), bar heights represent the mean, and error bars indicate the standard deviation across three independent runs with different random seeds. Results from individual runs are shown as dot points. Source data are provided as a Source Data file.

indicator of the generated data's usefulness. By leveraging this performance to inform the training process of the generation model, we ensure that the data produced is specifically optimized to improve the segmentation model. In previous methods, segmentation performance does not impact the process of data augmentation and generation. As a result, the augmented/generated data might not be effectively tailored for training the segmentation model. Furthermore, our framework learns a generative model that excels in generating data with greater diversity compared to existing augmentation methods.

GenSeg excels in surpassing semi-supervised segmentation methods without the need for external unlabeled images. In the context of medical imaging, collecting even unlabeled images presents a significant challenge due to stringent privacy concerns and regulatory constraints (e.g., IRB approval), thereby reducing the feasibility of semi-supervised methods. Despite the use of unlabeled real images, semi-supervised approaches underperform compared to GenSeg. This is primarily because these methods struggle to generate accurate masks for unlabeled images, meaning they are less effective at creating labeled training data. On the other hand, GenSeg is capable of producing high-quality images from masks, ensuring a close correspondence between the images' content and the masks, thereby efficiently generating labeled training examples.

Our framework is designed to be universally applicable and independent of specific models. This design choice enables it to augment the capabilities of a broad spectrum of semantic segmentation models. To apply our framework to a specific segmentation model, the only requirement is to integrate the segmentation model into the second and third stages of our framework. This straightforward process enables researchers and practitioners to easily utilize our approach to improve the performance of diverse semantic segmentation models.

GenSeg presents several limitations that warrant attention. First, although GenSeg generates high-quality synthetic image-mask pairs, its performance may still be dependent on the quality and diversity of the limited real-world training data available. If the small dataset used to guide the generation process is highly biased or unrepresentative, the synthetic data produced may inherit these biases, potentially leading to suboptimal generalization on unseen cases. Additionally, while GenSeg demonstrates strong OOD performance, its generalization capabilities may diminish when faced with divergent datasets or imaging modalities that differ significantly from the training set. Furthermore, although GenSeg does not require extensive unlabeled data like semi-supervised methods, it still relies on a small set of expert-annotated data to initiate the synthetic data generation process, meaning that its utility may be limited in cases where even a small annotated dataset is difficult to obtain. Finally, the integration of GenSeg into clinical workflows would require validation in real-world settings to ensure that the synthetic data does not introduce artifacts or inconsistencies that could affect diagnostic decisions. Addressing these limitations in future iterations of GenSeg would be crucial for broadening its applicability and improving its robustness in diverse clinical environments.

Future research on GenSeg can progress in multiple directions. A key area is improving synthetic data generation to better represent complex anatomical structures and the variability inherent in diverse imaging modalities. This could involve refining the MLO process to capture finer details or incorporating advanced neural architectures to enhance the quality of synthetic images. Additionally, using generative models that can learn from limited examples may help GenSeg generalize more effectively across a broader range of medical scenarios. Another important direction is applying domain adaptation techniques to improve GenSeg's robustness when encountering datasets that diverge significantly from the training data, such as novel imaging technologies or underrepresented patient populations. This would ensure more reliable performance in real-world clinical settings. Extending GenSeg's capabilities beyond segmentation to tackle other medical imaging challenges, like anomaly detection, image registration, or multimodal image fusion, could further expand its utility. Such developments would position GenSeg as a more versatile tool for medical image analysis, addressing a wider array of diagnostic and treatment planning needs. Furthermore, integrating feedback from clinical experts into the synthetic data generation process could increase its clinical relevance, aligning outputs more closely with diagnostic practices. These research directions could enhance GenSeg's adaptability and effectiveness across diverse medical imaging task.

An important consideration in evaluating the realism and utility of generated masks is how their variability compares to inter-reader variability observed in expert annotations. While our current study does not include a direct comparison—due to the use of datasets with only a single reference annotation per image—this is a valuable direction for future work. Qualitatively, we find that the augmented masks produced by our generative model exhibit anatomically plausible and semantically consistent variations, often resembling the natural diversity seen across patients and imaging conditions. Quantitatively, the consistent improvements in segmentation accuracy suggest that these synthetic masks enrich the training set with meaningful variability. Nevertheless, a systematic comparison with inter-reader variability would provide deeper insights into the clinical realism of the generated data. Incorporating multi-reader datasets in future evaluations could help assess whether the diversity introduced by generative augmentation aligns with the range of acceptable expert interpretations.

In summary, GenSeg is a robust data generation tool that seamlessly integrates with current semantic segmentation models. It significantly enhances both in-domain and OOD generalization performance in ultra low-data regimes, markedly boosting sample efficiency. Furthermore, it surpasses state-of-the-art methods in data augmentation and semi-supervised learning.

## Methods
### Overview of GenSeg
GenSeg consists of a data generation model and a medical image segmentation model. The data generation model is based on

conditional generative adversarial networks (GANs)[49,50]. It comprises two main components: a mask-to-image generator and a discriminator. Uniquely, our generator has a learnable neural architecture[51], as opposed to the fixed architecture commonly seen in previous GAN models. This generator, with weight parameters $\mathbf{G}$ and a learnable architecture $\mathbf{A}$, takes a segmentation mask as input and generates a corresponding medical image. The discriminator, with learnable weight parameters $\mathbf{H}$ and a fixed architecture, differentiates between synthetic and real medical images. The segmentation model has learnable weight parameters $\mathbf{S}$ and a fixed architecture.

Data generation is executed in a reverse manner. Starting with an expert-annotated segmentation mask $\mathbf{M}$, we first apply basic image augmentations, such as rotation, flipping, etc., to produce an augmented mask $\widehat{\mathbf{M}}$. This mask is then fed into the mask-to-image generator, resulting in a medical image $\widehat{\mathbf{I}}(\widehat{\mathbf{M}}, \mathbf{G}, \mathbf{A})$, which corresponds to $\widehat{\mathbf{M}}$, i.e., pixels in $\widehat{\mathbf{I}}(\widehat{\mathbf{M}}, \mathbf{G}, \mathbf{A})$ can be semantically labeled using $\widehat{\mathbf{M}}$. Each image-mask pair $(\widehat{\mathbf{I}}(\widehat{\mathbf{M}}, \mathbf{G}, \mathbf{A}), \widehat{\mathbf{M}})$ forms an augmented example for training the segmentation model. Like other deep learning-based segmentation methods, GenSeg has access to a training set comprised of real image-mask pairs $\mathcal{D}_{\text{seg}}^{\text{tr}} = \{\mathbf{I}_n^{(\text{tr})}, \mathbf{M}_n^{(\text{tr})}\}_{n=1}^{N_{\text{tr}}}$ and a validation set $\mathcal{D}_{\text{seg}}^{\text{val}} = \{\mathbf{I}_n^{(\text{val})}, \mathbf{M}_n^{(\text{val})}\}_{n=1}^{N_{\text{val}}}$.

### A multi-level optimization framework for GenSeg

GenSeg employs a MLO strategy across three distinct stages. The initial stage focuses on training the data generation model, where we fix the generator's architecture $\mathbf{A}$ and train the weight parameters of both the generator ($\mathbf{G}$) and the discriminator ($\mathbf{H}$). To facilitate this training, we modify the segmentation training dataset $\mathcal{D}_{\text{seg}}^{\text{tr}}$ by swapping the roles of inputs and outputs, resulting in a new dataset $\mathcal{D}_{\text{gan}} = \{\mathbf{M}_n^{(\text{tr})}, \mathbf{I}_n^{(\text{tr})}\}_{n=1}^{N_{\text{tr}}}$. In this setup, $\mathbf{M}_n^{(\text{tr})}$ serves as the input, while $\mathbf{I}_n^{(\text{tr})}$ acts as the output for our mask-to-image GAN model.

Let $L_{\text{gan}}$ represent the GAN training objective, a cross-entropy function that evaluates the discriminator's ability to distinguish between real and generated images. The discriminator's goal is to maximize $L_{\text{gan}}$, effectively separating real images from generated ones. Conversely, the generator strives to minimize $L_{\text{gan}}$, generating images that are so realistic they become indistinguishable from real ones. This process is encapsulated in the following minimax optimization problem:

$$\mathbf{G}^*(\mathbf{A}), \mathbf{H}^* = \text{argmin}_{\mathbf{G}} \ \text{argmax}_{\mathbf{H}} \ L_{\text{gan}}(\mathbf{G}, \mathbf{A}, \mathbf{H}, \mathcal{D}_{\text{gan}}), \quad (1)$$

where $\mathbf{G}^*(\mathbf{A})$ indicates that the optimally trained generator $\mathbf{G}^*$ is dependent on the architecture $\mathbf{A}$. This dependency arises because $\mathbf{G}^*$ is the outcome of optimizing the training objective function, which in turn is influenced by $\mathbf{A}$. $\mathbf{A}$ is tentatively fixed at this stage and will be updated later. Otherwise, if we learn $\mathbf{A}$ by minimizing the training loss $L_{\text{gan}}$, it may lead to a trivial solution characterized by an overly large and complex architecture. Such a solution would likely overfit the training data perfectly but perform poorly on unseen test data.

In the second stage, we leverage the trained generator to generate synthetic training examples using the aforementioned process, where expert-annotated masks are from $\mathcal{D}_{\text{seg}}^{\text{tr}}$. Let $\widehat{\mathcal{D}}(\mathbf{G}^*(\mathbf{A}), \mathcal{D}_{\text{seg}}^{\text{tr}})$ represent the generated data. We then use $\widehat{\mathcal{D}}(\mathbf{G}^*(\mathbf{A}), \mathcal{D}_{\text{seg}}^{\text{tr}})$ and real training data $\mathcal{D}_{\text{seg}}^{\text{tr}}$ to train the segmentation model $\mathbf{S}$ by minimizing a segmentation loss $L_{\text{seg}}$ (pixel-wise cross-entropy loss). This training is formulated as the following optimization problem:

$$\mathbf{S}^*(\mathbf{A}) = \text{argmin}_{\mathbf{S}} \ L_{\text{seg}}(\mathbf{S}, \widehat{\mathcal{D}}(\mathbf{G}^*(\mathbf{A}), \mathcal{D}_{\text{seg}}^{\text{tr}})) + \gamma L_{\text{seg}}(\mathbf{S}, \mathcal{D}_{\text{seg}}^{\text{tr}}), \quad (2)$$

where $\gamma$ is a trade-off parameter.

In the third stage, we assess the performance of the trained segmentation model on the validation dataset $\mathcal{D}_{\text{seg}}^{\text{val}}$. The validation loss,

$L_{\text{seg}}(\mathbf{S}^*(\mathbf{A}), \mathcal{D}_{\text{seg}}^{\text{val}})$, serves as an indicator of the quality of the generated data. If the generated data is of inferior quality, it will likely result in $\mathbf{S}^*(\mathbf{A})$−trained on this data - performing poorly on the validation set, reflected in a high validation loss. Thus, enhancing the quality of generated data can be achieved by minimizing $L_{\text{seg}}(\mathbf{S}^*(\mathbf{A}), \mathcal{D}_{\text{seg}}^{\text{val}})$ w.r.t the generator's architecture $\mathbf{A}$. This objective is encapsulated in the following optimization problem:

$$\text{min}_{\mathbf{A}} \ L_{\text{seg}}(\mathbf{S}^*(\mathbf{A}), \mathcal{D}_{\text{seg}}^{\text{val}}). \quad (3)$$

We can integrate these stages into a MLO problem as follows:

$$
\begin{aligned}
\text{min}_{\mathbf{A}} \quad & L_{\text{seg}}(\mathbf{S}^*(\mathbf{A}), \mathcal{D}_{\text{seg}}^{\text{val}}) \\
s.t \quad & \mathbf{S}^*(\mathbf{A}) = \text{argmin}_{\mathbf{S}} \ L_{\text{seg}}(\mathbf{S}, \widehat{\mathcal{D}}(\mathbf{G}^*(\mathbf{A}), \mathcal{D}_{\text{seg}}^{\text{tr}})) + \\
& \qquad\qquad\qquad \gamma L_{\text{seg}}(\mathbf{S}, \mathcal{D}_{\text{seg}}^{\text{tr}}) \\
& \mathbf{G}^*(\mathbf{A}), \mathbf{H}^* = \text{argmin}_{\mathbf{G}} \ \text{argmax}_{\mathbf{H}} \ L_{\text{gan}}(\mathbf{G}, \mathbf{A}, \mathbf{H}, \mathcal{D}_{\text{gan}})
\end{aligned}
\quad (4)
$$

In this formulation, the levels are interdependent. The output $\mathbf{G}^*(\mathbf{A})$ from the first level defines the objective for the second level, the output $\mathbf{S}^*(\mathbf{A})$ from the second level defines the objective for the third level, and the optimization variable $\mathbf{A}$ in the third level defines the objective function in the first level.

### Architecture search space

To enhance the generation of medical images by accurately capturing their distinctive characteristics, we make the generator's architecture searchable. Inspired by DARTS[51], we employ a differentiable search method that is not only computationally efficient but also allows for a flexible exploration of architectural designs. Our search space is structured as a series of computational cells, each forming a directed acyclic graph that includes an input node, an output node, and intermediate nodes comprising $K$ different operators, such as convolution and transposed convolution. These operators are each tied to a learnable selection weight, $\alpha$, ranging from 0 to 1, where a higher $\alpha$ value indicates a stronger preference for incorporating that operator into the final architecture. The process of architecture search is essentially the optimization of these selection weights. Let Conv-$xyz$ and UpConv-$xyz$ denote a convolution operator and a transposed convolution operator respectively, where $x$ represents the kernel size, $y$ the stride, and $z$ the padding. The pool of candidate operators includes Conv/UpConv-421, Conv/UpConv-622, and Conv/UpConv-823, i.e., the number of operators $K$ is 3. For any given cell $i$ with input $\mathbf{x}_i$, the output $\mathbf{y}_i$ is determined by the formula $\mathbf{y}_i = \sum_{k=1}^{K} \alpha_{i,k} o_{i,k}(\mathbf{x}_i)$, where $o_{i,k}$ represents the $k$-th operator in the cell, and $\alpha_{i,k}$ is its corresponding selection weight. Consequently, the architecture of the generator can be succinctly described by the set of all selection weights, denoted as $\mathbf{A} = \{\alpha_{i,k}\}$. Architecture search amounts to learning $\mathbf{A}$.

### Optimization algorithm

We develop a gradient-based method to solve the MLO problem in Eq. (4). First, we approximate $\mathbf{G}^*(\mathbf{A})$ using one-step gradient descent update of $\mathbf{G}$ w.r.t $L_{\text{gan}}(\mathbf{G}, \mathbf{A}, \mathbf{H}, \mathcal{D}_{\text{gan}})$:

$$\mathbf{G}^*(\mathbf{A}) \approx \mathbf{G}' = \mathbf{G} - \eta_{\text{g}} \nabla_G L_{\text{gan}}(\mathbf{G}, \mathbf{A}, \mathbf{H}, \mathcal{D}_{\text{gan}}), \quad (5)$$

where $\eta_{\text{g}}$ is a learning rate. Similarly, we approximate $\mathbf{H}^*$ using one-step gradient ascent update of $\mathbf{H}$ w.r.t $L_{\text{gan}}(\mathbf{G}, \mathbf{A}, \mathbf{H}, \mathcal{D}_{\text{gan}})$:

$$\mathbf{H}^* \approx \mathbf{H}' = \mathbf{H} + \eta_{\text{h}} \nabla_{\mathbf{H}} L_{\text{gan}}(\mathbf{G}, \mathbf{A}, \mathbf{H}, \mathcal{D}_{\text{gan}}). \quad (6)$$

Then we plug $\mathbf{G}^*(\mathbf{A}) \approx \mathbf{G}'$ into the objective function in the second level, yielding an approximated objective. We approximate $\mathbf{S}^*(\mathbf{A})$ using

one-step gradient ascent update of $\mathbf{S}$ w.r.t the approximated objective:

$$\mathbf{S}^*(\mathbf{A}) \approx \mathbf{S}' = \mathbf{S} - \eta_s \nabla_{\mathbf{S}} \left( L_{seg}(\mathbf{S}, \widehat{\mathcal{D}}(\mathbf{G}', \mathcal{D}_{seg}^{tr})) + \gamma L_{seg}(\mathbf{S}, \mathcal{D}_{seg}^{tr}) \right). \tag{7}$$

Finally, we plug $\mathbf{S}^*(\mathbf{A}) \approx \mathbf{S}'$ into the validation loss in the third level, yielding an approximated validation loss. We update $\mathbf{A}$ using gradient descent w.r.t the approximated loss:

$$\mathbf{A} \leftarrow \mathbf{A} - \eta_a \nabla_{\mathbf{A}} L_{seg}(\mathbf{S}', \mathcal{D}_{seg}^{val}). \tag{8}$$

After $\mathbf{A}$ is updated, we plug it into Eq. (5) to update $\mathbf{G}$ again. The update steps in Eq. (5)–(8) iterate until convergence.

The gradient $\nabla_{\mathbf{A}} L_{seg}(\mathbf{S}', \mathcal{D}_{seg}^{val})$ can be calculated as follows:

$$\nabla_{\mathbf{A}} L_{seg}(\mathbf{S}', \mathcal{D}_{seg}^{val}) = \frac{\partial \mathbf{G}'}{\partial \mathbf{A}} \frac{\partial \mathbf{S}'}{\partial \mathbf{G}'} \frac{\partial L_{seg}(\mathbf{S}', \mathcal{D}_{seg}^{val})}{\partial \mathbf{S}'}, \tag{9}$$

where

$$\frac{\partial \mathbf{G}'}{\partial \mathbf{A}} = -\eta_g \nabla_{\mathbf{A}, \mathbf{G}}^2 L_{gan}(\mathbf{G}, \mathbf{A}, \mathbf{H}, \mathcal{D}_{gan}), \tag{10}$$

$$\frac{\partial \mathbf{S}'}{\partial \mathbf{G}'} = -\eta_s \nabla_{\mathbf{G}', \mathbf{S}}^2 \left( L_{seg}(\mathbf{S}, \widehat{\mathcal{D}}(\mathbf{G}', \mathcal{D}_{seg}^{tr})) + \gamma L_{seg}(\mathbf{S}, \mathcal{D}_{seg}^{tr}) \right). \tag{11}$$

## Datasets

In this study, we focused on the segmentation of skin lesions from dermoscopy images, lungs from chest X-ray images, breast cancer from ultrasound images, placental vessels from fetoscopic images, polyps from colonoscopy images, foot ulcers from standard camera images, intraretinal cystoid fluid from OCT images, and left ventricle and myocardial wall from echocardiography images, utilizing 19 datasets. Additionally, we extended GenSeg to 3D image segmentation and evaluated its effectiveness on two 3D medical imaging datasets for hippocampus and liver segmentation. Each dataset was randomly partitioned into training, validation, and test sets, with the corresponding statistics presented in Supplementary Table 1. The number of training examples was determined based on two considerations. The first consideration is consistency with prior work. For well-established benchmarks such as ISIC, we adopted low-data configurations used in previous studies to enable fair comparisons. For example, in the skin lesion segmentation task, we followed the setup used in SemanticGAN[20]. The second consideration is dataset-specific complexity. For datasets without standardized low-sample training protocols, we selected training set sizes based on task difficulty. Specifically, datasets involving more complex anatomical structures, high intra-class variability, or low contrast typically required more training samples to obtain stable performance. In contrast, datasets with simpler and well-defined structures could be effectively learned using fewer samples.

For skin lesion segmentation from dermoscopy images, we utilized the ISIC2018[52], PH2[53], DermIS[54], and DermQuest[55] datasets. The ISIC2018 dataset, provided by the International Skin Imaging Collaboration (ISIC) 2018 Challenge, comprises 2,594 dermoscopy images, each meticulously annotated with pixel-level skin lesion labels. The PH2 dataset, acquired at the Dermatology Service of Hospital Pedro Hispano in Matosinhos, Portugal, contains 200 dermoscopic images of melanocytic lesions. These images are in 8-bit RGB color format with a resolution of $768 \times 560$ pixels. DermIS offers a comprehensive collection of dermatological images covering a range of skin conditions,

including dermatitis, psoriasis, eczema, and skin cancer. DermQuest includes 137 images representing two types of skin lesions: melanoma and nevus.

For lung segmentation from chest X-rays, we utilized the JSRT[56], NLM-MC[57], NLM-SZ[57], and COVID-QU-Ex[58] datasets. The JSRT dataset consists of 247 chest X-ray images from Japanese patients, each accompanied by manually annotated ground truth masks that delineate the lung regions. The NLM-MC dataset was collected from the Department of Health and Human Services in Montgomery County, Maryland, USA. It includes 138 frontal chest X-rays, with manual lung segmentations provided. Of these, 80 images represent normal cases, while 58 exhibit manifestations of tuberculosis (TB). The images are available in two resolutions: $4020 \times 4892$ pixels and $4892 \times 4020$ pixels. The NLM-SZ dataset, sourced from Shenzhen No.3 People's Hospital, Guangdong, China, contains 566 frontal chest X-rays in PNG format. Image sizes vary but are approximately $3000 \times 3000$ pixels. The COVID-QU-Ex dataset, compiled by researchers at Qatar University, comprises a large collection of chest X-ray images, including 11,956 COVID-19 cases, 11,263 non-COVID infections, and 10,701 normal instances. Ground-truth lung segmentation masks are provided for all images in this dataset.

For placental vessel segmentation from fetoscopic images, we utilized the FPD[59] and FetReg[60] datasets. The FPD dataset comprises 482 frames extracted from six distinct in vivo fetoscopic procedure videos. To reduce redundancy and ensure a diverse set of annotated samples, the videos were down-sampled from 25 to 1 fps, and each frame was resized to a resolution of $448 \times 448$ pixels. Each frame is provided with a corresponding segmentation mask that precisely outlines the blood vessels. The FetReg dataset, developed for the FetReg2021 challenge, is the first large-scale, multi-center dataset focused on fetoscopy laser photocoagulation procedures. It contains 2718 pixel-wise annotated images, categorizing background, vessel, fetus, and tool classes, sourced from 24 different in vivo TTTS fetoscopic surgeries.

For polyp segmentation from colonoscopic images, we utilized the KVASIR[61] and CVC-ClinicDB[62] datasets. Polyps are recognized as precursors to colorectal cancer and are detected in nearly half of individuals aged 50 and older who undergo screening colonoscopy, with their prevalence increasing with age. Early detection of polyps significantly improves survival rates from colorectal cancer. The KVASIR dataset was collected using endoscopic equipment at Vestre Viken Health Trust (VV) in Norway, which consists of four hospitals and provides healthcare services to a population of 470,000. The dataset includes images with varying resolutions, ranging from $720 \times 576$ to $1920 \times 1072$ pixels. It contains 1000 polyp images, each accompanied by a corresponding segmentation mask, with annotations verified by experienced endoscopists. CVC-ClinicDB comprises frames extracted from colonoscopy videos and consists of 612 images with a resolution of $384 \times 288$ pixels, derived from 31 colonoscopy sequences. videos.

For breast cancer segmentation, we utilized the BUID dataset[63], which consists of 630 breast ultrasound images collected from 600 female patients aged between 25 and 75 years. The images have an average resolution of $500 \times 500$ pixels. For foot ulcer segmentation, we utilized data from the FUSeg challenge[64], which includes over 1000 images collected over a span of two years from hundreds of patients. The raw images were captured using Canon SX 620 HS digital cameras and iPad Pro under uncontrolled lighting conditions, with diverse backgrounds. For the segmentation of intraretinal cystoids from Optical Coherence Tomography (OCT) images, we utilized the Intraretinal Cystoid Fluid (ICFluid) dataset[65]. This dataset comprises 1460 OCT images along with their corresponding masks for the Cystoid Macular Edema ocular condition. For the segmentation of left ventricles and myocardial wall, we employed data examples from the ETAB benchmark[66]. It is constructed from five publicly available echocardiogram datasets, encompassing diverse cohorts and providing echocardiographies with a variety of views and annotations.

For 3D medical image segmentation tasks, we utilized two datasets from the MSD challenge[4]: Task04 (hippocampus segmentation) and Task03 (liver segmentation). The hippocampus segmentation task focuses on segmenting the hippocampal region from single-modality MR images. The hippocampus is a key brain structure involved in memory formation, spatial navigation, and emotion processing. Anatomically, it is often divided into anterior and posterior regions, each associated with distinct cognitive and emotional functions. In our experiments, we merged the anterior and posterior regions into a single segmentation category. The dataset includes MR scans from 394 patients, officially split into 260 training and 131 test cases. Since test annotations are not publicly available, we split the original training set into training and test subsets using an 80:20 ratio. During training, the training set was further split into training and validation sets, also with an 80:20 ratio. The Task03 dataset for liver segmentation contains 201 contrast-enhanced CT scans from patients with primary liver cancers and metastatic disease originating from colorectal, breast, and lung cancers. Among these, 123 cases are officially designated for training. We applied the same data-splitting strategy as used in the hippocampus dataset, resulting in 98 training cases and 25 test cases.

## Metrics

For all segmentation tasks except skin lesion segmentation, we used the Dice score as the evaluation metric, adhering to established conventions in the field[67]. The Dice score is calculated as $\frac{2|A \cap B|}{|A| + |B|}$, where $A$ represents the algorithm's prediction and $B$ denotes the ground truth. For skin lesion segmentation, we followed the guidelines of the ISIC challenge[68] and employed the Jaccard index, also known as intersection-over-union (IoU), as the performance metric. The Jaccard index is computed as $\frac{|A \cap B|}{|A \cup B|}$ for each patient case. These metrics provide a robust assessment of the overlap between the predicted segmentation mask and the ground truth.

## Hyperparameters

In our method, mask augmentation was performed using a series of operations, including rotation, flipping, and translation, applied in a random sequence. The mask-to-image generation model was based on the Pix2Pix framework[50], with an architecture that was made searchable, as depicted in Fig. 1b. The tradeoff parameter $\gamma$ was set to 1. We configured the training process to perform 5000 iterations. The RMSprop optimizer[69] was utilized for training the segmentation model. It was set with an initial learning rate of $1e-5$, a momentum of 0.9, and a weight decay of $1e-3$. Additionally, the ReduceLROnPlateau scheduler was employed to dynamically adjust the learning rate according to the model's performance throughout the training period. Specifically, the scheduler was configured with a patience of 2 and set to max mode, meaning it monitored the model's validation performance and adjusted the learning rate to maximize validation accuracy. For training the mask-to-image generation model, the Adam optimizer[70] was chosen, configured with an initial learning rate of $1e-5$, beta values of (0.5, 0.999), and a weight decay of $1e-3$. Adam was also applied for optimizing the architecture variables, with a learning rate of $1e-4$, beta values of (0.5, 0.999), and weight decay of $1e-5$. At the end of each epoch, we assessed the performance of the trained segmentation model on a validation set. The model checkpoint with the best validation performance was selected as the final model. The experiments were conducted on A100 GPUs, with each method being run three times using randomly initialized model weights. We report the average results along with the standard deviation across these three runs.

## Reporting summary

Further information on research design is available in the Nature Portfolio Reporting Summary linked to this article.

## Data availability

The skin lesion segmentation data used in this study are available in the ISIC, PH2 [https://www.fc.up.pt/addi/ph2The lung segmentation data used in this study are available in the JSRT, COVID-QU-Ex [https://www.kaggle.com/datasets/anasmohammedtahir/covidqu], NLM-MC, and NLM-SZ [http://archive.nlm.nih.gov/repos/chestImages.php] databases. The breast cancer segmentation data used in this study are available in the BUID [https://www.kaggle.com/datasets/aryashah2k/breast-ultrasound-images-dataset?select=Dataset_BUSI_with_GT] database. The placental vessel segmentation data used in this study are available in the FPD [https://www.ucl.ac.uk/interventional-surgical-sciences/fetoscopy-placenta-data] and FetReg [https://www.ucl.ac.uk/interventional-surgical-sciences/weiss-open-research/weiss-open-data-server] databases. The polyp segmentation data used in this study are available in the KVASIR and CVC-Clinic [https://www.kaggle.com/datasets/balraj98/cvcclinicdb] databases. The foot ulcer segmentation data used in this study are available in the FUSeg [https://github.com/uwm-bigdata/wound-segmentation/tree/master] database. The intraretinal cystoid segmentation data used in this study are available in the ICFluid [https://www.kaggle.com/datasets/zeeshanahmed13/intraretinal-cystoid-fluid] database. The left ventricle and myocardial wall segmentation data used in this study are available in the ETAB database. The hippocampus and liver segmentation data used in this study are available in the MSD [https://drive.google.com/drive/folders/1HqEgzzS8BV2c7xYNrZdEAnrHk7osJJ--2] database. Source data are provided with this paper.

## Code availability

The source code used in this study is available at https://github.com/importZL/GenSeg and is archived at https://zenodo.org/records/15427671[71]. GenSeg is licensed under the Apache 2.0 License[72].

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

## Acknowledgements

P.X. acknowledges funding support from NSF IIS2405974, NSF IIS2339216, NIH R35GM157217, and NIH R21GM154171.

## Author contributions

L.Z. contributed to conceptualization, methodology, software, investigation, analysis, writing-original draft, and writing-editing. B.J. contributed to conceptualization, methodology, and writing-original draft. A.A., R.W., D.W., E.S., and J.Z. contributed to investigation, analysis, and writing-editing. P.X. contributed to conceptualization, methodology, investigation, analysis, writing-original draft, and writing-editing.

## Competing interests

E.S. is a paid consultant to Pheno.AI, Ltd. The other authors declare no competing interests.
