## [Peer Review File · Nature Communications]

Generative AI Enables Medical Image Segmentation in Ultra Low-Data Regimes

Corresponding Author: Professor Pengtao Xie

Version 0:

Reviewer comments:

Reviewer #2

(Remarks to the Author)

Zhang et al introduce a novel method using generative models for data augmentation in low-data regimes. The primary novelty in the GeneSeg method is encompassing the data generation and segmentation models in an end-to-end approach. As a result, GeneSeg can simultaneously generate synthetic images with corresponding segmentation masks and use them to train a segmentation model in a single algorithm. GeneSeg is validated on expertly annotated authentic images which enables legitimate performance assessment. Validation was performed across a variety of imaging sources with performance improvements over using segmentation models without a generative component. U-net and DeepLab were both evaluated as deep learning backbones. The paper is well-written and comprehensive. This is a well polished paper, as a result my comments are minor:

How are the error bars computed in Fig 2, Fig 4, Fig 5, Fig 6, Fig 7?

In Fig 5, does vanilla just mean that there is no data augmentation?

Is there a sense for how the variability in generated masks compare to inter-reader variability?

(Remarks on code availability)

Reviewer #3

(Remarks to the Author)

The authors introduce a novel generative-based augmentation technique aimed at improving segmentation tasks. The primary contribution of this work lies in the model's ability to simultaneously generate and augment segmentation data, whereas most state-of-the-art methods follow a two-step, independent approach. The authors propose a multi-level optimization strategy to enable end-to-end training of both the generative model and the segmentation model.

Strengths:

The writing is very clear, and the initial problem of data scarcity is well articulated in the introduction. The limitations of traditional two-step augmentation approaches are clearly identified. The authors propose an innovative solution that enables a simultaneous generation/augmentation approach, where the generative process is calibrated based on the segmentation results. The use of segmentation results as a feedback is a particularly interesting, as it could open new possibilities for future improvements in generative model-based augmentation. The evaluation of the method appears to be quite comprehensive, with a few exceptions that need to be addressed (see comments below). Overall, the methodology is robust and aligns with the expected standards of the field.

Nonetheless, this paper still contains some drawbacks that need to be addressed (or justified) before it can be considered for acceptance.

Drawbacks:

1. Firstly, the paper is well written; however, its structure and organization could be improved. The placement of figures should be optimized to improve clarity and visibility. Additionally, the Method section has been positioned after the references. Is this a deliberate choice?
2. The section on the generative model should be further developed. Based on the architecture presented in Figure 1, one might be led to believe that it is a simple UNet-like architecture. However, the method section mentions that it is a GAN-based generative model. The authors focus on micro-architectural details, such as highlighting the presence of an encoder-decoder structure, but they fail to mention the role of a discriminator, which, in my view, is a more critical component in a GAN-based generative model. It would be helpful to clarify this aspect, possibly by updating the figure caption to make it more explicit.
3. The use of learnable multi-branch convolutions is an interesting approach for adapting to the data. However, this adaptation is limited to the micro-architectural level, and its impact does not seem as significant as one might expect. In Extended Data Figure 2.c, the authors present a comparison of several GAN-based generative models. Does this comparison account for the multi-branch functionality? It would be beneficial to include an ablation study to quantify the specific impact of the learnable multi-branch convolutions.
4. The authors compare their method only with the traditional UNet and DeepLab models. Adding a third comparison method would be appropriate, particularly with nnUNet, as it is the current state-of-the-art in segmentation and is widely used in clinical settings. While the nnUNet framework may not be compatible with your augmentation approach (as GenSeg-nnUNet would be difficult to implement), it would still be interesting to observe how a standalone nnUNet performs in such ultra-low data regimes compared to your method.
5. It would also be more interesting to see generative models such as VAEs or Diffusion models in action, as their optimization framework is better suited for data simulation in lower data regimes. Coupled with the GenSeg approach, this could yield better results. It has already been demonstrated that VAE-based generative models, for instance, are more robust in data-scarce environments [1-3]. It may be worth considering a comparison with one of these methods, as a VAE is relatively easy to implement. Similarly, in Figure 5, your method is compared to WGAN. While WGAN might be slightly better at handling data scarcity than a standard GAN, it is still a GAN-based architecture, which remains very data-hungry. A comparison with other generative frameworks would be beneficial.
6. The generation is calibrated based on the segmentation results, which is both innovative and effective, and the augmentation is made in an end-to-end manner. However, the process still follows a two-step approach when it comes to generative images and masks. Some works have proposed the generating both simultaneously, which improves both the fidelity and flexibility of the generated image/mask pairs and better alignment of the mask to the region of interest [4]. Have you considered using a model capable of generating both the image and the mask simultaneously? In your specific case, the conditional GAN can synthesis a multi-channel mask/image pair.
7. Additionally to the previous point, the use of traditional augmentation on masks severely limits the range of possibilities and, from my perspective, may limit the capabilities of the mask-to-image generative model. The number of modes that can be covered by traditional augmentation techniques is limited, and moreover, certain operations cannot be applied to specific modalities, as they could destroy the semantic structure of the images. For these reasons, some approaches such as Elastic and Deformable augmentations have been proposed [5], have you considered comparing your method with this type of augmentations?
8. In Extended Data Figure 9, the augmentation applied to the masks of placental vessel images includes rotations. However, this type of image can sometimes be sensitive to the angle of the vessels. Don't you think that this could lead to aberrant generations?
9. How did you choose the size of the low-sample training set for each dataset? Is there any reasoning behind these choices?
10. It would also be interesting to compare the segmentation methods between the ultra-low data regime and the fully available data regime. In Figure 4, does the maximum number of training examples correspond to the maximum amount of available data? I believe this is not the case, given the dataset descriptions you provided. It would therefore be valuable to see what the results would have been if the entire available dataset had been used. Even if these results were better than GenSeg's, it could provide insight into the contribution of GenSeg and help quantify the value of the method.
11. The authors state: "Our method involved three-fold cross-validation on each dataset." Given that the training sets for each dataset were limited to small sizes ranging from 40 to 100 images (according to the Figure 2 caption), does each validation fold contain the same number of images? If so, the evaluation should be conducted on the entire set of available images that were not used during training.
12. The authors have already evaluated their method on a large number of datasets, which is notably impressive. However, all of these datasets are 2D, which lightly simplifies the problem. It would be interesting to introduce a 3D dataset, if possible, particularly from MRI or PET scans, as these data are multidimensional and more complex. Currently, data scarcity primarily affects these types of modalities, for example, brain cancer on MRIs, where there is a lack of sufficient data due to the very short lifespan of the patients.

13. The GenSeg method clearly improves the results across different evaluations, effectively demonstrating the contribution of the GenSeg approach. However, for certain datasets such as FetReg and ISIC, the results still seem unsatisfactory. A DICE score of 0.51 after improvement remains insufficient, especially considering that it is a binary segmentation task. Perhaps increasing the amount of training data would help improve the DICE score, shifting from an insufficient result without augmentation to a satisfactory result using GenSeg. Such results would be better to present in the paper. In my opinion, a result can be considered satisfactory starting at a DICE score of 70-75%.

- [1] - Chadebec, C., & Allasonnière, S. (2021). Data augmentation with variational autoencoders and manifold sampling. In Deep Generative Models, and Data Augmentation, Labelling, and Imperfections: First Workshop, DGM4MICCAI 2021, and First Workshop, DALI 2021, Held in Conjunction with MICCAI 2021, Strasbourg, France, October 1, 2021, Proceedings 1 (pp. 184-192). Springer International Publishing.
- [2] - Yin Dai, Fayu Liu, Weibing Chen, Yue Liu, Lifu Shi, Sheng Liu, Yuhang Zhou, et al. Swin mae: Masked autoencoders for small datasets. *Computers in Biology and Medicine*, 161:107037, 2023.
- [3] - Kebaili, A., Lapuyade-Lahorgue, J., Vera, P., & Ruan, S. (2024). Discriminative Hamiltonian Variational Autoencoder for Accurate Tumor Segmentation in Data-Scarce Regimes. arXiv preprint arXiv:2406.11659.
- [4] - Kebaili, A., Lapuyade-Lahorgue, J., Vera, P., & Ruan, S. (2024). 3D MRI Synthesis with Slice-Based Latent Diffusion Models: Improving Tumor Segmentation Tasks in Data-Scarce Regimes. arXiv preprint arXiv:2406.05421.
- [5] - Garcea, F., Serra, A., Lamberti, F., & Morra, L. (2023). Data augmentation for medical imaging: A systematic literature review. *Computers in Biology and Medicine*, 152, 106391.

(Remarks on code availability)

Version 1:

Reviewer comments:

Reviewer #2

(Remarks to the Author)
Wonderful work!

(Remarks on code availability)

Reviewer #3

(Remarks to the Author)
Thank you for your responses to my comments. I find them satisfactory and appreciate that you have incorporated the comparative studies I suggested. However, you should add the new references in your response letter (page 16) to your revised article.

(Remarks on code availability)

Response Letter for “Generative AI Enables Medical Image Segmentation in Ultra Low-Data Regimes”

Li Zhang, Basu Jindal, Ahmed Alaa, Robert Weinreb, David Wilson,
Eran Segal, James Zou, Pengtao Xie

We sincerely thank the reviewers for their time and effort in providing a thorough evaluation of our manuscript. We greatly appreciate their positive reviews, insightful comments, and constructive suggestions, which we have carefully addressed and leveraged to improve our work in the revised submission. In response to the reviewers’ feedback, we conducted several additional experiments and provided detailed, point-by-point replies to all comments. For clarity, each comment is presented in a boxed format, followed by our corresponding response. We also summarize the revisions made in response to each comment and indicate their locations in the revised manuscript. Changes in the manuscript are highlighted in blue. All relevant figures and tables are included in the updated manuscript. The corresponding code used during the revision is available at: <https://github.com/importZL/GenSeg>. We look forward to any further comments from the reviewers.

Contents

1 Responses to Reviewer #2	3
1.1 Paper summary	3
1.2 Clarify how error bars are computed	3
1.3 Clarify the definition of “vanilla”	3
1.4 Comparison of generated mask variability with inter-reader variability.	3
2 Responses to Reviewer #3	5
2.1 Paper summary and strengths	5
2.2 Improve paper structure and organization, figure placement, and positioning of references	6
2.3 Clarify GAN architecture	6
2.4 Ablation study on learnable multi-branch convolutions	6
2.5 Comparison with nnUNet	7
2.6 Comparison with other generative models including VAE and diffusion models	9
2.7 Ablation study on generating images and masks simultaneously	10
2.8 Comparison with elastic and deformable augmentations	11
2.9 Ablation study on rotation augmentation in placental vessel segmentation	12
2.10 Choice of low-Sample training set size	12
2.11 Comparison between ultra-low data regime and fully available data regime	13
2.12 Clarify “three-fold cross-validation”	14
2.13 Extension to 3D segmentation	14
2.14 Improve GenSeg’s performance on FetReg and ISIC datasets to 70-75%	15

1 Responses to Reviewer #2

We greatly appreciate the reviewer’s positive review and valuable feedback, which we have carefully addressed and leveraged to improve our work. In summary, we have made the following improvements to the manuscript based on the reviewer’s suggestions:

- Clarified how error bars are computed.
- Clarified the definition of “vanilla”.
- Addressed the question of how variability in generated masks compares to inter-reader variability.

Below, we provide point-by-point responses to each comment.

1.1 Paper summary

Zhang et al introduce a novel method using generative models for data augmentation in low-data regimes. The primary novelty in the GeneSeg method is encompassing the data generation and segmentation models in an end-to-end approach. As a result, GeneSeg can simultaneously generate synthetic images with corresponding segmentation masks and use them to train a segmentation model in a single algorithm. GeneSeg is validated on expertly annotated authentic images which enables legitimate performance assessment. Validation was performed across a variety of imaging sources with performance improvements over using segmentation models without a generative component. U-net and DeepLab were both evaluated as deep learning backbones. The paper is well-written and comprehensive. This is a well polished paper, as a result my comments are minor.

We sincerely thank the reviewer for the positive and encouraging feedback. We are grateful for the recognition of our method’s novelty in integrating data generation and segmentation into an end-to-end framework. We also appreciate the reviewer’s comments that the manuscript is well written, comprehensive, and well polished.

1.2 Clarify how error bars are computed

How are the error bars computed in Fig 2, Fig 4, Fig 5, Fig 6, Fig 7?

We thank the reviewer for the question. Each experiment was run three times using different random seeds. The error bars shown in the figures represent the standard deviation computed across these three independent runs. We have added this clarification to the caption of Fig. 2 in the revised manuscript.

1.3 Clarify the definition of “vanilla”

In Fig 5, does vanilla just mean that there is no data augmentation?

We appreciate the reviewer’s question. Yes, in Figure 5, the term “vanilla” refers to training the model without any data augmentation. This serves as a baseline to help isolate and evaluate the contributions of various augmentation strategies. We have clarified this definition in the caption of Fig. 5 in the revised manuscript.

1.4 Comparison of generated mask variability with inter-reader variability.

Is there a sense for how the variability in generated masks compares to inter-reader variability?

We thank the reviewer for raising this important point. In the current study, we did not explicitly benchmark the variability of our generated masks against annotations from multiple human readers, as most of the datasets used provide only a single expert-labeled ground truth. We agree that such a comparison would provide valuable insights into the realism and diversity of the generated masks and could serve as an additional metric for evaluating generative models in medical image segmentation.

Our method aims to generate augmented masks that are anatomically plausible and semantically consistent with the original structures. Qualitatively, we observe that the diversity in the generated masks often reflects natural variations seen across patient populations and imaging conditions. Quantitatively, the improvements in segmentation performance suggest that these generated masks provide meaningful and complementary information to the original training data.

In future work, we plan to incorporate multi-reader datasets to systematically assess how the variability introduced by our generative model compares to inter-reader variability, which could further validate the clinical relevance of the generated data.

We have added this discussion to the “Discussion” section of the revised manuscript.

2 Responses to Reviewer #3

We greatly appreciate the reviewer’s positive review and valuable feedback, which we have carefully addressed and leveraged to improve our work. In summary, we have made the following improvements to the manuscript based on the reviewer’s suggestions:

- Improved paper structure and organization, figure placement, and positioning of references.
- Clarified that our method utilizes a GAN-based model for data generation.
- Demonstrated the effectiveness of learnable multi-branch convolutions through ablation studies.
- Compared GenSeg with nnUNet where GenSeg-UNet outperforms nnUNet with a large margin.
- Compared GAN-based data generation with other generative models, including VAE and diffusion models, and found that the GAN-based approach remains a competitive choice in terms of both segmentation performance and computational efficiency.
- Performed ablation studies comparing simultaneous generation of images and masks with the two-step approach, where masks are generated first followed by image generation. The results demonstrate that the two-step approach outperforms simultaneous generation.
- Investigated elastic augmentations, finding that elastic augmentation slightly improves results when combined with standard augmentations but degrades performance when used alone.
- Investigated the impact of rotation augmentation in placental vessel segmentation and found that applying small rotations yields better performance than omitting rotation.
- Explained the rationale behind the choice of low-sample training set sizes.
- Compared model performance under both ultra-low and fully available data regimes, demonstrating that GenSeg remains effective even when trained with the full dataset.
- Clarified “three-fold cross-validation”.
- Extended GenSeg to 3D segmentation tasks, demonstrating its effectiveness in improving performance for 3D medical image segmentation.
- Improved the performance of GenSeg on ISIC and FetReg datasets to 70–75%.

Below, we provide point-by-point responses to each comment.

2.1 Paper summary and strengths

The authors introduce a novel generative-based augmentation technique aimed at improving segmentation tasks. The primary contribution of this work lies in the model’s ability to simultaneously generate and augment segmentation data, whereas most state-of-the-art methods follow a two-step, independent approach. The authors propose a multi-level optimization strategy to enable end-to-end training of both the generative model and the segmentation model.

Strengths: The writing is very clear, and the initial problem of data scarcity is well articulated in the introduction. The limitations of traditional two-step augmentation approaches are clearly identified. The authors propose an innovative solution that enables a simultaneous generation/augmentation approach, where the generative process is calibrated based on the segmentation results. The use of segmentation results as a feedback is a particularly interesting, as it could open new possibilities for future improvements in generative model-based augmentation. The evaluation of the method appears to be quite comprehensive, with a few exceptions that need to be addressed (see comments below). Overall, the methodology is robust and aligns with the expected standards of the field.

We sincerely thank the reviewer for the positive and encouraging feedback, including the recognition that our method is novel, interesting, and robust, that the evaluation is quite comprehensive, and that the writing is very clear.

2.2 Improve paper structure and organization, figure placement, and positioning of references

Firstly, the paper is well written; however, its structure and organization could be improved. The placement of figures should be optimized to improve clarity and visibility. Additionally, the Method section has been positioned after the references. Is this a deliberate choice?

We thank the reviewer for the positive comment that the paper is well written, as well as for the helpful suggestions regarding structure and organization. In response, we have adjusted the placement of figures so that they appear close to where they are first mentioned in the text, improving clarity and readability. In the original version, we followed the convention used in some Nature Portfolio journals, where the Methods section appears after the references cited in the Introduction, Results, and Discussion sections. In this revision, we have reorganized the manuscript to place all references at the end, following the Methods section. Additionally, we have improved the overall structure by moving certain subsections from the Methods section to the Results section to enhance logical flow and coherence.

2.3 Clarify GAN architecture

The section on the generative model should be further developed. Based on the architecture presented in Figure 1, one might be led to believe that it is a simple UNet-like architecture. However, the method section mentions that it is a GAN-based generative model. The authors focus on micro-architectural details, such as highlighting the presence of an encoder-decoder structure, but they fail to mention the role of a discriminator, which, in my view, is a more critical component in a GAN-based generative model. It would be helpful to clarify this aspect, possibly by updating the figure caption to make it more explicit.

We appreciate the reviewer’s valuable feedback. To address this, we have revised Figure 1 to explicitly depict the full architecture of our GAN-based framework, including the discriminator. We have also updated the figure caption to clearly state that the generative model is GAN-based. Additionally, we have expanded the description of the generative model in the “GenSeg overview” subsection of the Results section to clarify that it is a GAN-based model consisting of both a generator and a discriminator.

2.4 Ablation study on learnable multi-branch convolutions

The use of learnable multi-branch convolutions is an interesting approach for adapting to the data. However, this adaptation is limited to the micro-architectural level, and its impact does not seem as significant as one might expect. In Extended Data Figure 2.c, the authors present a comparison of several GAN-based generative models. Does this comparison account for the multi-branch functionality? It would be beneficial to include an ablation study to quantify the specific impact of the learnable multi-branch convolutions.

Response Figure 1: Ablation study on learnable multi-branch convolutions, with UNet as the segmentation model.

We thank the reviewer for the positive comment that the use of learnable multi-branch convolutions is an interesting approach, as well as for the valuable suggestion to further quantify its impact. We first confirm that in the results presented in the original Extended Data Figure 2c (Figure 9d in the revised paper), where several GAN-based models were evaluated as mask-to-image

Response Table 1: Two-sided t-test results comparing single-branch and multi-branch convolutions.

Method	Dataset	P-value	Effect size	T-statistic
Learnable Multi-branch	ISIC-200	0.016	3.462	4.239
vs	ICFluid-50	0.007	6.566	8.042
Single-branch	BUID-100	0.001	8.121	9.946
Fixed Multi-branch	ISIC-200	0.128	1.603	1.964
vs	ICFluid-50	0.002	7.401	9.065
Single-branch	BUID-100	0.038	2.745	3.362
Learnable Multi-branch	ISIC-200	0.051	2.249	2.755
vs	ICFluid-50	0.037	3.709	4.544
Fixed Multi-branch	BUID-100	0.032	3.021	3.699

generators, we applied the multi-branch convolutional modification to the generator architectures of all models to ensure a fair and consistent comparison.

To directly quantify the impact of the multi-branch design, we conducted an ablation study involving three settings. In the first setting (Single-branch), we trained a standard single-branch Pix2Pix generator to synthesize images, which were then used to train the segmentation model in a separate stage. In the second setting (Fixed Multi-branch), we used a multi-branch Pix2Pix generator with fixed branch weights (i.e., all weights α in Figure 1c of the main paper are set to 1.), also trained independently from the segmentation model. In the third setting (Learnable Multi-branch), which corresponds to our full GenSeg framework, the generator was integrated into an end-to-end pipeline, where the branch weights α were learned by minimizing segmentation loss on the validation set.

We evaluated all three configurations on three representative tasks: skin lesion segmentation (ISIC dataset, 200 training examples), intraretinal cystoid segmentation (ICFluid dataset, 50 training examples), and breast cancer segmentation (BUID dataset, 100 training examples). As shown in Response Figure 1, the Fixed Multi-branch model consistently outperformed the Single-branch model, demonstrating the advantage of using multi-branch convolutions. Moreover, the Learnable Multi-branch model further improved performance, highlighting the benefit of learning the branch weights in a task-adaptive manner.

We attribute these improvements to the increased representational capacity of the multi-branch architecture, which enables the generator to learn a more diverse set of features tailored to varying spatial and structural characteristics across datasets. While the fixed multi-branch design provides architectural flexibility, the learnable version further strengthens performance by enabling end-to-end optimization that aligns synthetic data generation with the segmentation objective.

To assess the statistical significance of these improvements, we performed two-sided t-tests across the three tasks. As shown in Response Table 1, most p-values are below 0.05, confirming that the observed performance gains from the multi-branch architecture - particularly the learnable variant - are statistically significant.

In summary, this ablation study demonstrates that learnable multi-branch convolutions significantly improve segmentation accuracy, demonstrating their role as an important micro-architectural component of the GenSeg framework.

These updated results and corresponding analysis have been incorporated into the ‘‘Ablation study on learnable multi-branch convolutions’’ subsection of the Results section, Fig. 10e, and Extended Data Table 1 in the revised manuscript.

2.5 Comparison with nnUNet

The authors compare their method only with the traditional UNet and DeepLab models. Adding a third comparison method would be appropriate, particularly with nnUNet, as it is the current state-of-the-art in segmentation and is widely used in clinical settings. While the nnUNet framework may not be compatible with your augmentation approach (as GenSeg-nnUNet would be difficult to implement), it would still be interesting to observe how a standalone nnUNet performs in such ultra-low data regimes compared to your method.

We thank the reviewer for this insightful suggestion. Following the reviewer’s recommendation, we conducted additional experiments to evaluate the performance of a standalone nnUNet in the ultra-low data regimes considered in our study. We trained and evaluated nnUNet under both in-domain and out-of-domain settings across multiple segmentation tasks. To ensure robustness,

Response Figure 2: GenSeg-UNet consistently outperforms nnUNet across a range of segmentation tasks under in-domain scenarios.

Response Figure 3: GenSeg-UNet consistently demonstrates superior performance to nnUNet across diverse segmentation tasks in out-of-domain settings. In the X-Y notation, X refers to the training dataset and Y to the test dataset, where X and Y are from distinct distributions.

each experiment was repeated three times with different random seeds, and we report the averaged results along with standard deviations. As shown in Response Figures 2 and 3, GenSeg-UNet consistently outperforms nnUNet in these data-scarce settings. In in-domain scenarios (Response Figure 2), GenSeg-UNet achieves 1–7% (absolute percentages) higher performance scores across all tasks. In out-of-domain evaluations (Response Figure 3), which involve more substantial distributional shifts, GenSeg-UNet demonstrates even greater improvements across all tasks, outperforming nnUNet by 5–16% (absolute percentages). For instance, in the lung segmentation task, when trained on only 175 examples from the JSRT dataset and evaluated on the SZ dataset, GenSeg-UNet achieves a Dice score of 94.5%, compared to 78.4% with nnUNet - a substantial gain of 16.1%.

The superior performance of GenSeg over nnUNet in ultra-low data regimes can be attributed to fundamental differences in their augmentation strategies. nnUNet employs standard augmentation techniques such as rotation, scaling, Gaussian blur, and intensity adjustments, which, while effective in moderate- to large-scale data settings, offer limited diversity and adaptability in severely data-constrained scenarios. In contrast, GenSeg trains a deep generative model that synthesizes diverse and semantically consistent image-mask pairs tailored to the specific task and dataset. This generative augmentation approach introduces significantly greater variability into the training data, enabling the segmentation model to learn more robust and generalizable representations. By aligning the data generation process with segmentation performance through end-to-end multi-level optimization, GenSeg ensures that the synthesized data is not only realistic but also highly informative for improving downstream segmentation accuracy.

We have incorporated these new comparisons and the corresponding analysis into the revised manuscript, specifically in the subsections “GenSeg outperforms nnUNet across both in-domain and out-of-domain scenarios” of the Results section, Fig. 8a, and Fig. 8b.

2.6 Comparison with other generative models including VAE and diffusion models

It would also be more interesting to see generative models such as VAEs or Diffusion models in action, as their optimization framework is better suited for data simulation in lower data regimes. Coupled with the GenSeg approach, this could yield better results. It has already been demonstrated that VAE-based generative models, for instance, are more robust in data-scarce environments [4, 6, 10]. It may be worth considering a comparison with one of these methods, as a VAE is relatively easy to implement. Similarly, in Figure 5, your method is compared to WGAN. While WGAN might be slightly better at handling data scarcity than a standard GAN, it is still a GAN-based architecture, which remains very data-hungry. A comparison with other generative frameworks would be beneficial.

Response Figure 4: Ablation study evaluating the effectiveness of different generative models - including Pix2Pix (GAN-based), BBDM (diffusion-based), and Soft-Intro VAE (VAE-based) - under separate and end-to-end training strategies. Evaluations were conducted under both in-domain (a) and out-of-domain (b) scenarios, using UNet as the segmentation model. For out-of-domain scenarios, datasets are labeled in the format X-Y, where X denotes the training dataset and Y denotes the test dataset.

Response Figure 5: Comparison of training time (left) and model size (right) for Pix2Pix, BBDM, and Soft-Intro VAE within our end-to-end training framework, in skin lesion segmentation with 40 training examples from the ISIC dataset when using UNet as the segmentation model.

We appreciate the reviewer’s insightful suggestion. To address this point, we conducted additional experiments using two state-of-the-art generative models: Soft-Intro VAE [7], a representative VAE-based method, and BBDM [11], a diffusion-based generative model. We integrated each model into our GenSeg framework by using them to replace the original Pix2Pix mask-to-image generator, and evaluated their performance under both in-domain and out-of-domain scenarios. To maintain consistency with our architecture design, we adapted both BBDM and Soft-Intro VAE by incorporating our multi-branch convolutional cells into their generator networks. We trained the models using two strategies: (1) Separate, where the generative model is trained independently and then fixed before segmentation model training, and (2) End2End, our proposed multi-level optimization framework.

As shown in Response Figures 4a (in-domain) and 4b (out-of-domain), BBDM (End2End) achieved the highest performance across all datasets. The performance of Pix2Pix (End2End) and Soft-Intro VAE (End2End) was comparable, with both trailing slightly behind BBDM. However, as

illustrated in Response Figure 5, BBDM incurs significantly higher computational cost and model size compared to both Pix2Pix and Soft-Intro VAE under the End2End strategy. Considering the trade-off between segmentation performance and computational efficiency, Pix2Pix remains a practical and effective choice for our setting, particularly when computational resources are limited. Furthermore, all three End2End approaches consistently outperformed their respective Separate counterparts, highlighting the advantage of jointly optimizing the generative and segmentation models within an end-to-end training framework. This result reinforces the central premise of GenSeg: that aligning the data generation process with downstream segmentation performance leads to more effective learning.

These new results and the corresponding analysis have been incorporated into the revised manuscript, specifically in the ‘‘Ablation study evaluating different mask-to-image generative models’’ subsection of the Results section, Fig. 9a, Fig. 9b, and Fig. 9c.

2.7 Ablation study on generating images and masks simultaneously

The generation is calibrated based on the segmentation results, which is both innovative and effective, and the augmentation is made in an end-to-end manner. However, the process still follows a two-step approach when it comes to generative images and masks. Some works have proposed the generating both simultaneously, which improves both the fidelity and flexibility of the generated image/mask pairs and better alignment of the mask to the region of interest [9]. Have you considered using a model capable of generating both the image and the mask simultaneously? In your specific case, the conditional GAN can synthesis a multi-channel mask/image pair.

Response Figure 6: Ablation study comparing simultaneous image–mask generation with the two-step approach, where masks are first augmented and then used to generate images. The two-step strategy outperforms simultaneous generation. Experiments were conducted under both in-domain (a) and out-of-domain (b) settings. In out-of-domain scenarios, datasets are denoted in the format X–Y, where X represents the training dataset and Y the test dataset. UNet was used as the segmentation backbone in all experiments.

We thank the reviewer for the encouraging feedback regarding the innovation and effectiveness of our generation process and its end-to-end integration with segmentation. In response to the suggestion regarding simultaneous generation of images and masks, we conducted additional experiments using a generative model that produces both components jointly. While the work by Kebaili et al. [9] explores this idea in the context of 3D volume generation, its methodology does not directly align with our 2D image-based segmentation tasks. Therefore, we adopted the WGAN-GP model [2] for our investigation. In this setting, WGAN-GP takes a random noise vector sampled from a Gaussian distribution as input and simultaneously produces a medical image and its corresponding mask. To maintain consistency with our architectural design, we modified the original WGAN-GP by replacing its convolutional layers with our multi-branch convolutional cells. We then trained the model within our end-to-end optimization framework and compared its performance against our existing two-step approach, which first augments segmentation masks and then generates corresponding images conditioned on those masks.

As shown in Response Figure 6, our two-step approach consistently outperforms the WGAN-GP-based simultaneous generation method in both in-domain and out-of-domain settings. Notably, in the out-of-domain evaluations - where 40 examples from ISIC were used for training and PH2 and DermIS served as test sets - our method achieved 12.1% and 8.9% higher performance, respectively. The superior performance of the two-step approach over the simultaneous generation method can

be attributed to the explicit conditioning and structural alignment enforced during the data generation process. In our two-step pipeline, segmentation masks are first augmented and then used as conditioning inputs to guide the image generation process. This explicit conditioning enables the generative model (e.g., Pix2Pix) to synthesize images that are tightly aligned with the structural boundaries and semantics defined by the input mask. As a result, the generated image-mask pairs exhibit high spatial coherence and fidelity, which is crucial for effective segmentation model training. In contrast, the simultaneous generation approach, as implemented with WGAN-GP, synthesizes both the image and the mask jointly without enforcing a strong pixel-wise correspondence between the two outputs. This lack of explicit conditioning can lead to weaker structural alignment, especially in low-data regimes where the model may struggle to learn accurate joint representations. Specifically, it does not impose semantic constraints that guarantee the generated masks accurately delineate regions of interest within the corresponding images. This misalignment can reduce the effectiveness of the generated data in training downstream segmentation models.

These new results and the corresponding analysis have been incorporated into the revised manuscript, specifically in the ‘‘Ablation study investigating the impact of generating images and masks jointly’’ subsection of the Results section and in Fig. 9e and 9f.

2.8 Comparison with elastic and deformable augmentations

Additionally to the previous point, the use of traditional augmentation on masks severely limits the range of possibilities and, from my perspective, may limit the capabilities of the mask-to-image generative model. The number of modes that can be covered by traditional augmentation techniques is limited, and moreover, certain operations cannot be applied to specific modalities, as they could destroy the semantic structure of the images. For these reasons, some approaches such as Elastic and Deformable augmentations have been proposed [8], have you considered comparing your method with this type of augmentations?

Response Figure 7: Ablation study evaluating the impact of elastic augmentation under (a) in-domain and (b) out-of-domain settings. In out-of-domain scenarios, datasets are denoted in the format X-Y, where X represents the training dataset and Y the test dataset. UNet was used as the segmentation model.

We thank the reviewer for the insightful suggestion. In response, we conducted an ablation study to assess the impact of incorporating elastic augmentation into our training pipeline. Specifically, we compared the following three ablation settings:

- **Without Elastic** – using only our original set of augmentations (e.g., flipping, rotation, translation).
- **With Elastic** – combining our original augmentations with elastic augmentation.
- **Only Elastic** – using elastic augmentation alone, without any other augmentations.

As shown in Response Figures 7a and 7b, the combination of elastic and traditional augmentations (With Elastic) resulted in modest performance improvements across both in-domain and out-of-domain settings. However, the Without Elastic setting - using only our original traditional augmentations - consistently outperformed the Only Elastic setting, which applies elastic deformation alone, across all tasks. One possible explanation is that elastic augmentation, when used in isolation, may result in a narrower range of transformations, focusing primarily on localized shape distortions. While such deformations can be beneficial in mimicking anatomical variability, they may not capture broader appearance and geometric changes - such as orientation, scale, or

intensity shifts - that traditional augmentations introduce. As a result, relying solely on elastic transformations might limit the diversity of the training data and reduce generalization. These results suggest that traditional augmentations provide a strong and versatile baseline, and that combining them with elastic augmentations may offer additional benefits depending on the dataset characteristics and task requirements.

These new results and the corresponding analysis have been incorporated into the revised manuscript, specifically in the ‘‘Ablation study on elastic and deformable augmentations’’ subsection of the Results section, Fig. 10b, and Fig. 10c.

2.9 Ablation study on rotation augmentation in placental vessel segmentation

In Extended Data Figure 9, the augmentation applied to the masks of placental vessel images includes rotations. However, this type of image can sometimes be sensitive to the angle of the vessels. Don’t you think that this could lead to aberrant generations?

Response Figure 8: Ablation study evaluating the impact of rotation augmentation on placental vessel segmentation using the FetReg and FPD datasets with UNet as the segmentation model.

We thank the reviewer for raising this important question. To investigate this issue, we conducted additional experiments on two vessel segmentation datasets: FetReg and FPD, each using 100 training examples. We tested the impact of different degrees of rotation augmentation by comparing five settings: no rotation, small-angle rotation (-5° to 5°), moderate rotation (-15° to 15°), large rotation (-30° to 30°), and very large rotation (-45° to 45°).

As shown in Response Figure 8, on the FPD dataset, all degrees of rotation yielded better performance than the no-rotation baseline. On the FetReg dataset, small-angle rotation (-5° to 5°) provided the best performance, while increasing the rotation range gradually led to performance degradation. These observations align with the reviewer’s thought: large-angle rotations can distort vessel morphology and interfere with fine-grained structural cues essential for accurate segmentation, particularly in tasks requiring high spatial precision.

On the other hand, small-angle rotations appear beneficial. They introduce controlled variability that helps improve model generalization without compromising anatomical integrity. We hypothesize that such mild transformations encourage robustness to minor viewpoint changes while still preserving the spatial structure of vessels - an important consideration in vascular imaging.

In summary, our results confirm that vessel segmentation tasks are sensitive to large rotational transformations, which can negatively impact performance. However, mild rotations in the range of -5° to 5° strike a balance between augmentation diversity and structural preservation, leading to improved outcomes.

These new results and corresponding analysis have been incorporated into the revised manuscript, specifically in the ‘‘Ablation study on the impact of rotation augmentation in placental vessel segmentation’’ subsection of the Results section and Fig. 10d.

2.10 Choice of low-Sample training set size

How did you choose the size of the low-sample training set for each dataset? Is there any reasoning behind these choices?

We thank the reviewer for raising this important question. In our study, the number of training examples was determined based on the following two considerations:

- **Consistency with prior work:** For well-established benchmarks such as ISIC, we adopted low-data configurations used in previous studies to enable fair comparisons. For example, in the skin lesion segmentation task, we followed the setup used in SemanticGAN [12].
- **Dataset-specific complexity:** For datasets without standardized low-sample training protocols, we selected training set sizes based on task difficulty. Specifically, datasets involving more complex anatomical structures, high intra-class variability, or low contrast typically required more training samples to obtain stable performance. In contrast, datasets with simpler and well-defined structures could be effectively learned using fewer samples.

Overall, the choice of training set size reflects a balance between comparability with prior work and adaptation to dataset-specific characteristics. This clarification has been added to the revised manuscript, specifically in the “Datasets” subsection of the Methods section.

2.11 Comparison between ultra-low data regime and fully available data regime

It would also be interesting to compare the segmentation methods between the ultra-low data regime and the fully available data regime. In Figure 4, does the maximum number of training examples correspond to the maximum amount of available data? I believe this is not the case, given the dataset descriptions you provided. It would therefore be valuable to see what the results would have been if the entire available dataset had been used. Even if these results were better than GenSeg’s, it could provide insight into the contribution of GenSeg and help quantify the value of the method.

Response Figure 9: Comparison of model performance in ultra-low and full data regimes.

We thank the reviewer for this thoughtful suggestion. In response, we conducted additional experiments using the full available training sets. Specifically, we evaluated two training regimes for each segmentation method: (1) UNet-low and GenSeg-UNet-low, trained under the ultra-low data setting, and (2) UNet-full and GenSeg-UNet-full, trained using the full available training data.

As shown in Response Figure 9, we observe several findings. First, as expected, performance improves for all models as the training set size increases. Second, although models trained on the full dataset achieve the highest overall performance, the gap between GenSeg-UNet-low and UNet-full is relatively small in many cases. This suggests that GenSeg-UNet can deliver competitive performance even when trained on significantly less data. Third, GenSeg-UNet-full outperforms UNet-full when trained on the full dataset, showing that GenSeg’s generative augmentation strategy remains beneficial beyond the ultra-low data regime. These results highlight the versatility and effectiveness of the GenSeg framework across data availability conditions. Its ability to consistently enhance segmentation performance - regardless of dataset size - underscores the value of integrating generative augmentation within an end-to-end, task-driven learning paradigm. While GenSeg is particularly advantageous in scenarios with limited annotations, it also brings improvements in more data-rich settings by enriching the training signal and improving model generalization.

These new results and the corresponding analysis have been incorporated into the revised manuscript, specifically in the “GenSeg is effective in high-data regimes as well” subsection of the Results section and in Fig. 8e.

2.12 Clarify “three-fold cross-validation”

The authors state: “Our method involved three-fold cross-validation on each dataset.” Given that the training sets for each dataset were limited to small sizes ranging from 40 to 100 images (according to the Figure 2 caption), does each validation fold contain the same number of images? If so, the evaluation should be conducted on the entire set of available images that were not used during training.

We apologize for the confusion and the incorrect use of the term “cross-validation”. In our experiments, we performed three independent runs for each dataset using different random seeds, and then calculated the mean and standard deviation of the performance results from these runs. We realized that this should not be called “cross-validation” and have corrected the terminology in the revised manuscript.

2.13 Extension to 3D segmentation

The authors have already evaluated their method on a large number of datasets, which is notably impressive. However, all of these datasets are 2D, which lightly simplifies the problem. It would be interesting to introduce a 3D dataset, if possible, particularly from MRI or PET scans, as these data are multidimensional and more complex. Currently, data scarcity primarily affects these types of modalities, for example, brain cancer on MRIs, where there is a lack of sufficient data due to the very short lifespan of the patients.

Response Figure 10: Extension of the GenSeg framework to 3D medical image segmentation tasks under different training data regimes. “Hippo.-low” refers to training with an ultra-low data setting for hippocampus segmentation, while “Hippo.-full” refers to training with the full available dataset. The same settings are applied to the liver segmentation task.

We appreciate the reviewer’s comment that the evaluation of our method on a large number of datasets is notably impressive. We thank the reviewer for the thoughtful suggestion regarding the extension to 3D segmentation. In response, we extended our GenSeg framework to support 3D medical image segmentation by incorporating 3D UNet [5] as the segmentation model and Pix2PixNifTI [3] as the generative model, enabling joint generation and segmentation in a 3D volumetric setting. We make the architecture of the Pix2PixNifTI model searchable by replacing the convolution and transposed convolution layers in the original generator with our differentiable convolutional and transposed convolutional cells. The architecture parameters of the modified Pix2PixNifTI model are optimized by minimizing the segmentation loss on the validation set within our multi-level optimization-based framework. During training, the input 3D masks are first augmented using rotation and flipping transformations, and the generator then synthesizes 3D volumes from these augmented masks.

We evaluated this 3D extension on two datasets from the Medical Segmentation Decathlon (MSD) challenge [1], focusing on hippocampus and liver segmentation tasks. Experiments were conducted under both ultra-low data (40 training volumes) and full data settings (208 training volumes for hippocampus and 98 for liver). As shown in Response Figure 10, GenSeg consistently improved segmentation performance over the baseline 3D UNet in both regimes. Notably, in the ultra-low data setting, GenSeg yielded substantial gains, demonstrating its robustness and effectiveness in data-constrained 3D segmentation tasks. These results confirm that GenSeg generalizes beyond 2D segmentation and remains effective when applied to more complex 3D volumetric data.

These new results and the corresponding analysis have been incorporated into the revised manuscript, specifically in the “GenSeg enables 3D medical image segmentation” subsection of the

Results section and in Fig. 8d.

2.14 Improve GenSeg’s performance on FetReg and ISIC datasets to 70-75%

The GenSeg method clearly improves the results across different evaluations, effectively demonstrating the contribution of the GenSeg approach. However, for certain datasets such as FetReg and ISIC, the results still seem unsatisfactory. A DICE score of 0.51 after improvement remains insufficient, especially considering that it is a binary segmentation task. Perhaps increasing the amount of training data would help improve the DICE score, shifting from an insufficient result without augmentation to a satisfactory result using GenSeg. Such results would be better to present in the paper. In my opinion, a result can be considered satisfactory starting at a DICE score of 70-75%.

Response Figure 11: Further improvements on ISIC and FetReg datasets.

We thank the reviewer for the positive comment that GenSeg clearly improves results across different evaluations and effectively demonstrates its contribution. We also appreciate the thoughtful suggestion to further improve performance scores, particularly for challenging datasets such as FetReg and ISIC, with the goal of reaching a satisfactory performance range of 70–75%. To address this, we conducted additional experiments applying a combination of strategies: increasing the amount of training data, using a more proper segmentation backbone, and refining the augmentation settings.

As shown in Response Figure 11, we compare the performance of the original GenSeg method (“Original”) and the enhanced version (“Improved”) on both datasets. For the ISIC dataset (UNet was used as the segmentation model), increasing the training set size from 40 to 1000 examples resulted in a Jaccard score improvement from 67.3% to 73.9%, which falls within the satisfactory range suggested by the reviewer. For the FetReg dataset, reaching a Dice score above 70% proved more difficult due to the complexity of placental vessel segmentation, including high anatomical variability, low image contrast, and intricate vessel structures. To overcome these challenges, we narrowed the rotation augmentation range to $(-5^\circ$ to $5^\circ)$, used DeepLab as the segmentation model, and increased the number of training examples from 50 to 2000. These combined changes led to a substantial improvement, achieving a Dice score of 71.7%.

These results demonstrate that GenSeg, when paired with appropriate training data volume and model refinements, can achieve high segmentation performance even on difficult tasks.

These new results and corresponding analysis have been incorporated into the revised manuscript, specifically in the “Further improvement on ISIC and FetReg datasets” subsection of the Results section and Fig. 8f.

References

- [1] Michela Antonelli, Annika Reinke, Spyridon Bakas, Keyvan Farahani, Annette Kopp-Schneider, Bennett A Landman, Geert Litjens, Bjoern Menze, Olaf Ronneberger, Ronald M Summers, et al. The medical segmentation decathlon. *Nature communications*, 13(1):4128, 2022.
- [2] Martin Arjovsky, Soumith Chintala, and Léon Bottou. Wasserstein generative adversarial networks. In *International conference on machine learning*, pages 214–223. PMLR, 2017.
- [3] Giulia Baldini, Melanie Schmidt, Charlotte Zäske, and Liliana L Caldeira. Mri scan synthesis methods based on clustering and pix2pix. In *International Conference on Artificial Intelligence in Medicine*, pages 109–125. Springer, 2024.
- [4] Clément Chadebec and Stéphanie Allasonnière. Data augmentation with variational autoencoders and manifold sampling. In *MICCAI Workshop on Deep Generative Models*, pages 184–192. Springer, 2021.
- [5] Özgün Çiçek, Ahmed Abdulkadir, Soeren S Lienkamp, Thomas Brox, and Olaf Ronneberger. 3d u-net: learning dense volumetric segmentation from sparse annotation. In *Medical Image Computing and Computer-Assisted Intervention—MICCAI 2016: 19th International Conference, Athens, Greece, October 17-21, 2016, Proceedings, Part II 19*, pages 424–432. Springer, 2016.
- [6] Yin Dai, Fayu Liu, Weibing Chen, Yue Liu, Lifu Shi, Sheng Liu, Yuhang Zhou, et al. Swin mae: masked autoencoders for small datasets. *Computers in biology and medicine*, 161:107037, 2023.
- [7] Tal Daniel and Aviv Tamar. Soft-introvae: Analyzing and improving the introspective variational autoencoder. In *Proceedings of the IEEE/CVF Conference on Computer Vision and Pattern Recognition*, pages 4391–4400, 2021.
- [8] Fabio Garcea, Alessio Serra, Fabrizio Lamberti, and Lia Morra. Data augmentation for medical imaging: A systematic literature review. *Computers in Biology and Medicine*, 152:106391, 2023.
- [9] Aghiles Kebaili, Jérôme Lapuyade-Lahorgue, Pierre Vera, and Su Ruan. 3d mri synthesis with slice-based latent diffusion models: Improving tumor segmentation tasks in data-scarce regimes. In *2024 IEEE International Symposium on Biomedical Imaging (ISBI)*, pages 1–5. IEEE, 2024.
- [10] Aghiles Kebaili, Jérôme Lapuyade-Lahorgue, Pierre Vera, and Su Ruan. Discriminative hamiltonian variational autoencoder for accurate tumor segmentation in data-scarce regimes. *Neurocomputing*, 606:128360, 2024.
- [11] Bo Li, Kaitao Xue, Bin Liu, and Yu-Kun Lai. Bbdm: Image-to-image translation with brownian bridge diffusion models. In *Proceedings of the IEEE/CVF conference on computer vision and pattern Recognition*, pages 1952–1961, 2023.
- [12] Daiqing Li, Junlin Yang, Karsten Kreis, Antonio Torralba, and Sanja Fidler. Semantic segmentation with generative models: Semi-supervised learning and strong out-of-domain generalization. In *Proceedings of the IEEE/CVF conference on computer vision and pattern recognition*, pages 8300–8311, 2021.

Response Letter for “Generative AI Enables Medical Image Segmentation in Ultra Low-Data Regimes”

Li Zhang, Basu Jindal, Ahmed Alaa, Robert Weinreb, David Wilson,
Eran Segal, James Zou, Pengtao Xie

We sincerely thank the reviewers for their positive feedback and constructive suggestions, which we have carefully addressed and used to improve our work in the revised submission. We have also thoroughly addressed the editorial requests. Changes made in response to the editorial requests and the remaining reviewer comments are highlighted in orange in the manuscript.

Responses to Reviewer #2

Wonderful work!

We sincerely thank the reviewer for the positive and encouraging assessment of our revised manuscript and are grateful for the supportive remarks.

Responses to Reviewer #3

Thank you for your responses to my comments. I find them satisfactory and appreciate that you have incorporated the comparative studies I suggested. However, you should add the new references in your response letter (page 16) to your revised article.

We thank the reviewer for the positive assessment and for confirming satisfaction with our previous revisions. We appreciate the comment highlighting the need to add the new references (previously detailed on page 16 of our prior response letter) to the main article. These references have now been incorporated into the main reference list of the revised manuscript.